# STAG2 loss in Ewing sarcoma alters enhancer-promoter contacts dependent and independent of EWS::FLI1

Daniel Giménez-Llorente (ID), Ana Cuadrado (ID) ✉, María José Andreu, Inmaculada Sanclemente-Alamán (ID), Maria Solé-Ferran, Miriam Rodríguez-Corsino & Ana Losada (ID) ✉

## Abstract

Cohesin complexes carrying STAG1 or STAG2 organize the genome into chromatin loops. *STAG2* loss-of-function mutations promote metastasis in Ewing sarcoma, a pediatric cancer driven by the fusion transcription factor EWS::FLI1. We integrated transcriptomic data from patients and cellular models to identify a STAG2-dependent gene signature associated with worse prognosis. Subsequent genomic profiling and high-resolution chromatin interaction data from Capture Hi-C indicated that cohesin-STAG2 facilitates communication between EWS::FLI1-bound long GGAA repeats, presumably acting as neoenhancers, and their target promoters. Changes in CTCF-dependent chromatin contacts involving signature genes, unrelated to EWS::FLI1 binding, were also identified. STAG1 is unable to compensate for STAG2 loss and chromatin-bound cohesin is severely decreased, while levels of the processivity factor NIPBL remain unchanged, likely affecting DNA looping dynamics. These results illuminate how STAG2 loss modifies the chromatin interactome of Ewing sarcoma cells and provide a list of potential biomarkers and therapeutic targets.

**Keywords** Cohesin; Genome Organization; Transcription; Pediatric Cancer
**Subject Categories** Cancer; Chromatin, Transcription & Genomics

## Introduction

Ewing sarcoma is a bone cancer that affects both children and young adults (Grünewald et al, 2018). It is driven by a translocation that fuses the prion-like domain of a FET family RNA-binding protein with the DNA binding domain of an ETS transcription factor (Delattre et al, 1992). The most common chromosomal translocation, (11;22) (q24;q12), results in the EWS::FLI1 oncoprotein, which rewires the cell transcriptome to drive transformation (Riggi et al, 2008). EWS::FLI1 has been proposed to act as a pioneer transcription factor that induces de novo formation of active enhancers at GGAA repeats, thereby enabling the transcription of otherwise silent genes (Gangwal et al, 2008; Riggi et al, 2014; Boulay et al, 2018; Showpnil et al, 2022; Tomazou et al, 2015). Additional proteins cooperate with the oncogene, including the BAF remodeling complex, LSD1 demethylase, and the Polycomb Repressor Complex (PRC)1 component RING1B (Boulay et al, 2017; Theisen et al, 2021; Sánchez-Molina et al, 2020). Knock down (KD) of EWS::FLI1 in Ewing sarcoma cellular models also causes upregulation of several genes, but the mechanisms of EWS::FLI1-mediated repression are less clear. One hypothesis is that the oncoprotein displaces ETS factors from their canonical binding at single GGAA motifs (Riggi et al, 2014). The involvement of Polycomb has also been suggested, in this case through increased expression of the PRC2 component EZH2, which is responsible for H3K27 trimethylation and is a downstream target of EWS::FLI1 (Richter et al, 2009).

Ewing sarcoma is a highly aggressive cancer with a dismal prognosis in patients who present with metastasis at the time of diagnosis. Even in patients with localized disease, approximately 25% of cases do not respond well to treatment and frequently relapse (Riggi et al, 2021). To identify secondary genetic lesions that might dictate the more aggressive behavior of a fraction of tumors, genome sequencing detected loss-of-function mutations in the *STAG2* gene in 15–20% of patients. Importantly, these mutations are commonly associated with metastatic disease and poor survival outcomes (Brohl et al, 2014; Crompton et al, 2014; Tirode et al, 2014). STAG2 is one of two paralogs that are part of cohesin in somatic cells, the other being STAG1, together with SMC1A, SMC3, and RAD21 (Cuadrado and Losada, 2020). This complex organizes the genome into chromatin loops and mediates sister chromatid cohesion. Loss of function mutations in *STAG2*, an X-linked gene, are tolerated because cohesin-STAG1 is sufficient to maintain cell viability (van der Lelij et al, 2017; Liu et al, 2018). In contrast, embryonic development is impaired in the absence of either STAG1 or STAG2 (Remeseiro et al, 2012a; De Koninck et al, 2020). Both complexes fold the genome by loop extrusion in association with NIPBL and make progressively longer loops until they are stopped by CTCF or released from chromatin by PDS5-WAPL (Davidson, Peters, 2021; de Wit, Nora, 2022). However, cohesin-STAG1 is more important for topologically associated domain (TAD) boundary demarcation, whereas a more dynamic cohesin-STAG2 mediates local intra-TAD chromatin contacts such as those

Chromosome Dynamics Group, Molecular Oncology Programme, Spanish National Cancer Research Centre (CNIO), Melchor Fernández Almagro 3, 28029 Madrid, Spain.
✉E-mail: acuadrado@cnio.es; alosada@cnio.es

connecting enhancers and promoters (Kojic et al, 2018; Viny et al, 2019; Ochi et al, 2020; Richart et al, 2021; Van Der Weide et al, 2021; Cuadrado et al, 2019; Casa et al, 2020). The two cohesin variants display specific features in terms of chromatin association dynamics that most likely dictate their functional specificities (Wutz et al, 2020; Cuadrado et al, 2022; Alonso-Gil et al, 2023). Cohesin-STAG1 is found at CTCF-bound sites and is preferentially acetylated by ESCO1 in G1, which contributes to its longer chromatin residence time (Wutz et al, 2020). Cohesin interacts with WAPL and CTCF through a surface formed by RAD21 and STAG1/2 (Li et al, 2020) but for reasons that remain unclear, cohesin-STAG1 interacts more extensively with CTCF, whereas cohesin-STAG2 interacts more with WAPL (Wutz et al, 2020; Kojic et al, 2018; Cuadrado et al, 2022). The ratio of cohesin-STAG1 and cohesin-STAG2 in different cell types or developmental stages and the abundance of cohesin regulators such as WAPL or NIPBL affect gene expression and other genome processes that depend on cohesin-mediated loop formation (Losada et al, 2000; Kojic et al, 2018; Cuadrado et al, 2019; Kiefer et al, 2023; Luppino et al, 2022; Hill et al, 2020; Nakato et al, 2023). Thus, complete loss of STAG2 may alter the relative amount of cohesin and its regulators and impact loop extrusion dynamics, with consequences for the transcriptome. A previous study showed reduced CTCF-anchored loop extrusion in STAG2 deficient Ewing sarcoma cells based on CTCF HiChIP data (Surdez et al, 2021). How this reduction affects cis-interactions between EWS::FLI1-bound enhancers and target promoters could not be inferred from the genomic data. Nevertheless, many targets of the oncoprotein become downregulated after STAG2 loss thus resembling the so called EWS::FLI1 "low state" that promotes metastatic behavior by enhancing mesenchymal properties while reducing proliferation (Surdez et al, 2021; Adane et al, 2021; Franzetti et al, 2017). A concomitant study also suggested that STAG2 loss disrupts PRC2-regulated developmental programs (Adane et al, 2021).

We have taken a different approach to further understand the consequences of STAG2 loss in Ewing sarcoma. In the current study, we integrated gene expression data from patients and cellular models of Ewing sarcoma to identify a STAG2-dependent gene signature associated to worse prognosis. We used genomic profiling and high-resolution chromatin interaction data from Promoter Capture Hi-C (PCHi-C) to explore EWS::FLI1-dependent and independent mechanisms associated with these gene expression changes. We conclude that cohesin-STAG2 facilitates communication between EWS::FLI1-bound long GGAA repeats and their target promoters. Changes in CTCF-dependent chromatin contacts between promoters and distal regions unrelated to EWS::FLI1 binding may also contribute to a more aggressive phenotype. STAG1 was unable to compensate for the loss of STAG2 in STAG2 deficient cells and the total amount of cohesin on chromatin was decreased. Importantly, the ratio of NIPBL:cohesin increased, providing a plausible explanation for longer loops and possibly altered cohesin dynamics.

# Results

## Transcriptome changes in response to STAG2 loss in Ewing sarcoma patients

To better understand the consequences of STAG2 loss on the physiology of Ewing Sarcoma tumors, we analyzed the transcriptomes of patients with and without STAG2 mutations. These were obtained

from publicly available RNA-seq data of a cohort of 49 patients, of which 8 carried loss-of-function mutations in STAG2, as identified by whole genome sequencing (Tirode et al, 2014; Data ref: Tirode et al, 2014). Principal Component Analysis (PCA) revealed a subset of 10 cases with no detectable STAG2 mutations (labeled as WT*) that cluster with STAG2 mutant (MUT) cases (Fig. 1A; see Methods). Patients with STAG2 WT* tumors had worse prognosis than the rest with non-mutant or wild-type (WT) STAG2 (Fig. 1B) while their STAG2 mRNA levels were not necessarily lower (Fig. 1C). These data are in agreement with a recent study reporting that the frequency of STAG2 protein loss in Ewing sarcoma samples analyzed by immunohistochemistry is significantly higher than the 15–20% mutation rate identified by genomic sequencing (Shulman et al, 2022) and correlates with adverse prognosis (Shulman et al, 2022). Single-sample Gene Set Enrichment Analysis (ssGSEA), an extension of GSEA that calculates separate enrichment scores for each sample-gene set independently of phenotype/genotype labeling, confirmed the similarity between the transcriptomes of STAG2 WT* and MUT patients (Fig. 1D). It also showed significant enrichment of STAG2 MUT transcriptomes in gene sets related to pediatric cancers other than Ewing sarcoma, to metastasis and invasion, as well as to known functions of the cohesin complex, such as stemness, DNA replication/repair, and genome stability.

## A STAG2-dependent gene signature that predicts worse prognosis in Ewing sarcoma patients

We next obtained gene expression data for several clones of the Ewing sarcoma cell line A673, with and without STAG2, generated by CRISPR editing and verified by sequencing and by immunoblotting with antibodies directed against the amino and carboxi-terminal regions of the protein (Fig. EV1A,B). For some comparisons shown in this study, we also generated STAG1 deficient clones in the same cell line (Fig. EV1B). Loss of STAG2 resulted in the deregulation of 3814 genes (FDR < 0.05), of which 1800 were upregulated and 2014 were downregulated (Dataset EV1A). STAG2-dependent differential expression obtained from the comparison of patients from the aforementioned cohort (STAG2 WT and MUT) and A673 cells (parental and clones) yielded 232 differentially expressed genes (DEGs) in addition to STAG2, most of which were downregulated in the absence of STAG2 (Fig. 2A; Dataset EV1B,C). The expression levels of these genes were comparable in A673 cells knocked out (KO) and knocked down (KD) for STAG2, and changes were also observed in two other Ewing sarcoma cell lines, SK-N-MC and A4573, upon STAG2 loss and STAG2 restoration, respectively (Figs. 2A and EV1C,D). The expression of these genes in STAG2 WT* patients resembled that of STAG2 MUT better than STAG2 WT patients. Given the lower survival of STAG2 WT* and STAG2 MUT cases compared to STAG2 WT cases, it is likely that at least some of these gene expression changes are associated with more aggressive disease. To corroborate this hypothesis, we analyzed transcriptomic data from primary tumors of two additional independent cohorts with associated survival information (Savola et al, 2011; Data ref: Savola et al, 2011; Volchenboum et al, 2015; Data ref: Volchenboum et al, 2015). Patients were separated into two groups according to the expression of 232 genes from the STAG2-dependent gene signature (Fig. 2B; see Methods). More importantly, the signature predicted worse outcomes for patients with signature-like gene expression (Figs. 2C and EV2), which also showed lower STAG2 mRNA levels (Fig. 2D). Out of the 232 STAG2-dependent genes, only 55 are targets

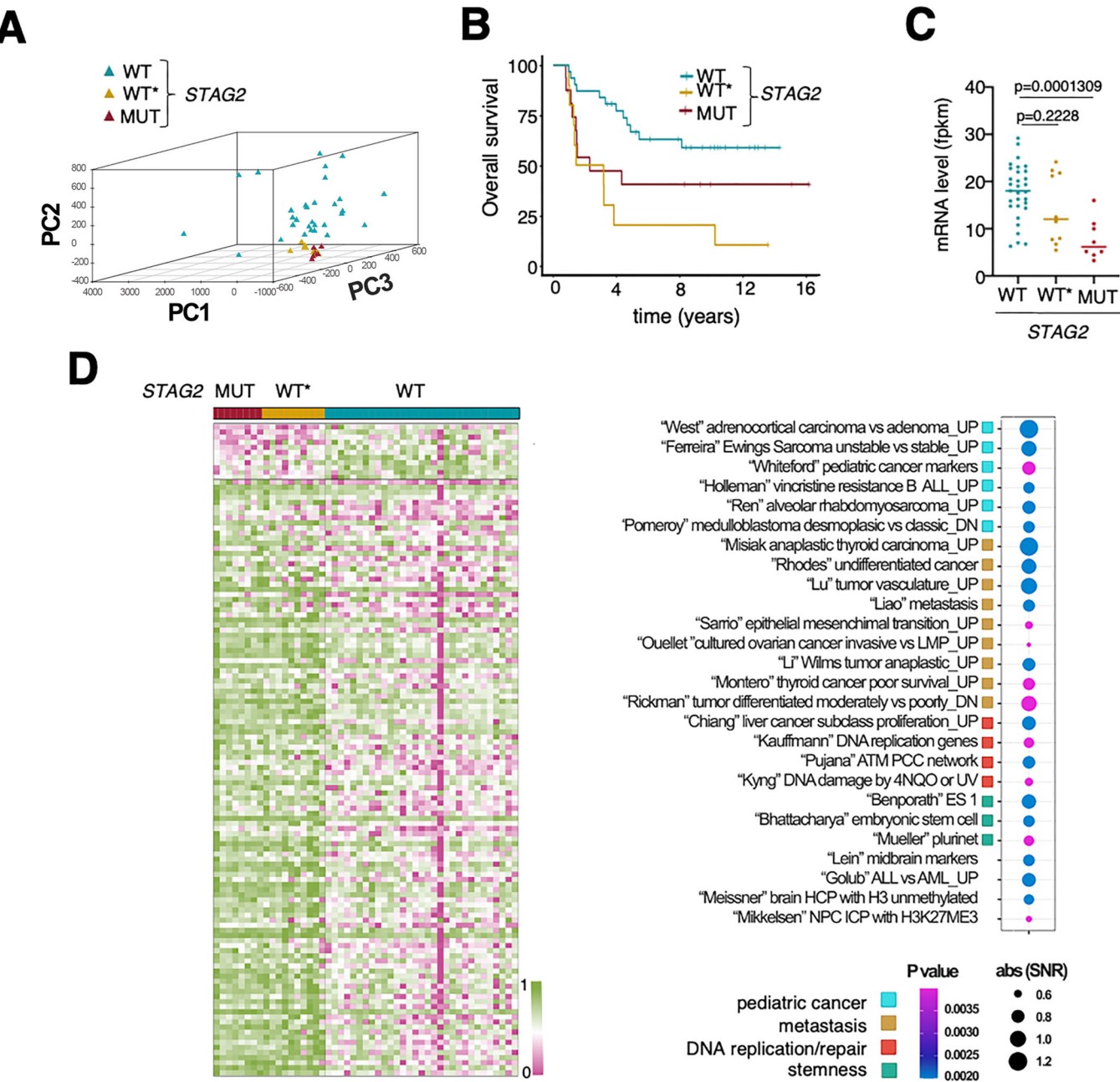

**Figure 1. Transcriptome changes in response to STAG2 loss in Ewing sarcoma patients.**

(A) PCA of transcriptome data from patients of the IGC cohort ($n = 49$) uncovers a subset of 10 patients (WT*) with no detectable STAG2 mutations that cluster with STAG2 mutant (MUT) cases. (B) Survival probability in the three groups of patients defined above. P values were obtained with Cox proportional hazards regression model comparing STAG2 WT* vs WT (0.0001309) and STAG2 WT* vs MUT (0.19806). (C) STAG2 expression levels in the three groups. Each dot corresponds to a patient, colored as in (A); p values were calculated with a Mann–Whitney test. (D) Left, ssGSEA separates STAG2 WT and MUT patients and further shows that STAG2 MUT and WT* patients display similar enrichments. The actual enrichment (Signal to Noise Ratio, SNR) and significance (P value) of selected gene sets in the transcriptome of STAG2 MUT patients were obtained using the tool "ComparativeMarkerSelection" from GenePattern and are shown on the right. Source data are available online for this figure.

of EWS::FLI1, that is, they become deregulated after oncogene KD (shadowed in Dataset EV1C). We conclude that STAG2 loss in Ewing sarcoma leads to transcriptional changes that extend beyond EWS::FLI1 target genes and that most likely contribute to adverse prognosis.

## Cohesin-STAG2 contributes to the establishment of EWS::FLI1 mediated interactions at long GGAA repeats

Hi-C analyses have shown that EWS::FLI1-centered chromatin loops generate highly connected interaction hubs that are

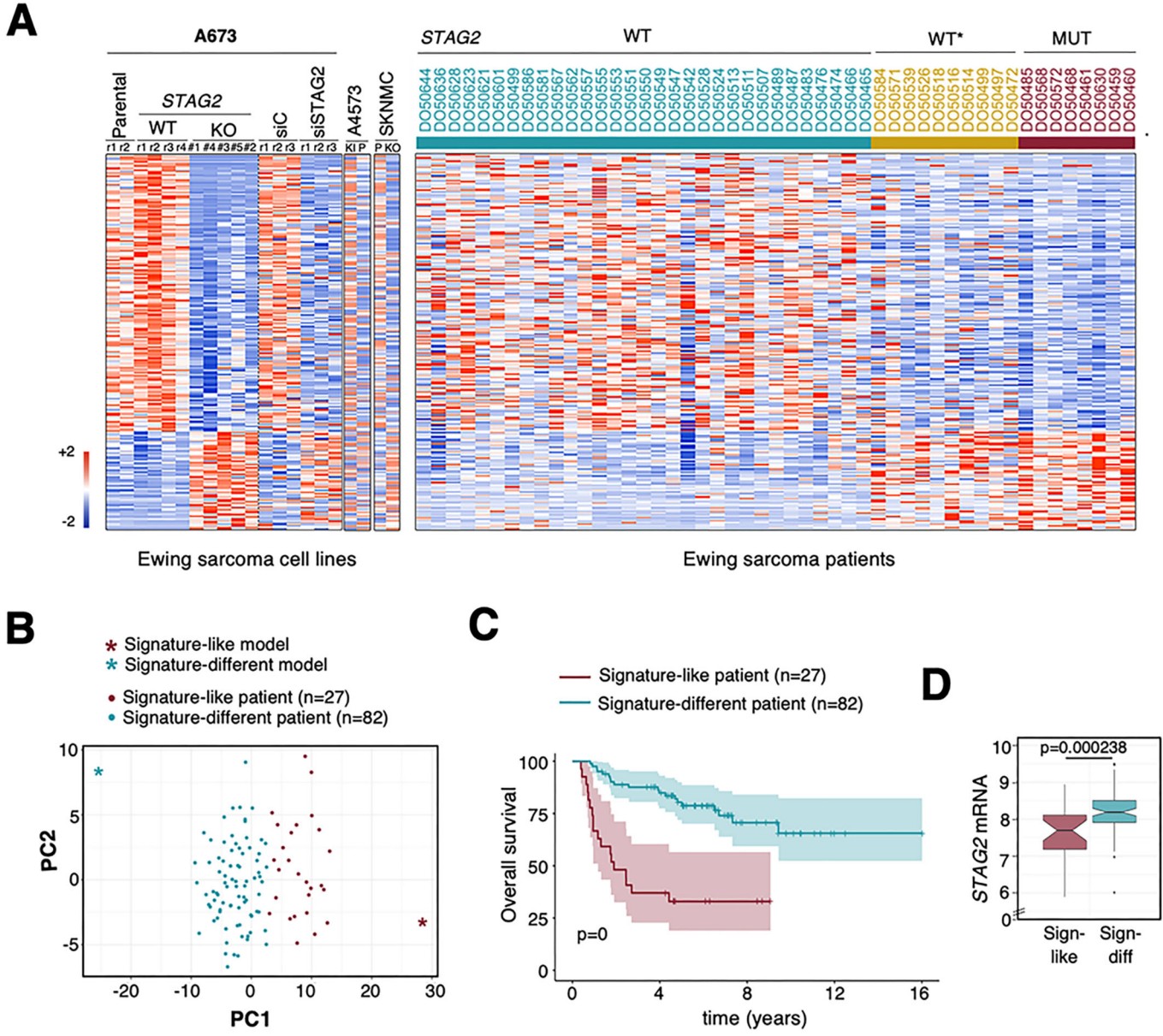

**Figure 2. Identification of a STAG2-dependent gene signature associated with survival in Ewing sarcoma.**

(A) Heatmap showing expression data for *STAG2* and 232 STAG2-dependent genes in cells and patients proficient or deficient for STAG2, as indicated. Patient data from the IGC cohort (Data ref: Tirode et al, 2014). Replicates (r) were obtained for A673 parental cell line (P), WT clone and for parental clones either mock transfected (siC) or transfected with siSTAG2. See Dataset EV1 for lists of DEGs in each condition. (B) PCA segregates patients from two independent cohorts (*n* = 109; primary tumors only; Data ref: Savola et al, 2011; Data ref: Volchenboum et al, 2015) according to expression of 232 STAG2-dependent genes. See Methods for details. (C) Overall survival probability in these patients (*n* = 27 signature-like and *n* = 82 signature-different). *P* value calculated with Cox proportional hazards regression model. (D) mRNA levels of *STAG2* in the same patients. *P* value calculated with Mann–Whitney test. Source data are available online for this figure.

important for the oncogenic transcriptional program (Sanalkumar et al, 2023; Showpnil et al, 2022). Given the role of cohesin in 3D genome organization, we reckoned that cohesin-STAG2 may facilitate the establishment of these interactions. To test this possibility, we analyzed the genomic distribution of EWS::FLI1, cohesin, H3K27ac, p300, and CTCF using available ChIP-seq data from different Ewing sarcoma cell lines (Table EV1). A heatmap of the EWS::FLI1 peaks ordered according to the number of GGAA repeats they encompass shows that the oncoprotein binds more

strongly to long (*n* > 4) GGAA repeats, also considered microsatellites, and this binding is essential for their proposed activity as enhancers (Fig. EV3A). Indeed, EWS::FLI1 downregulation in A673 and SK-N-MC cells dramatically reduced H3K27ac and p300 occupancy at these sites; the opposite occurred when the oncogene was introduced into mesenchymal stem cells (Fig. EV3A). Changes in loci with short (up to four) GGAA repeats were much less pronounced. Importantly, STAG2 loss reduced EWS::FLI1 binding to the short GGAA repeats but increased its presence at long

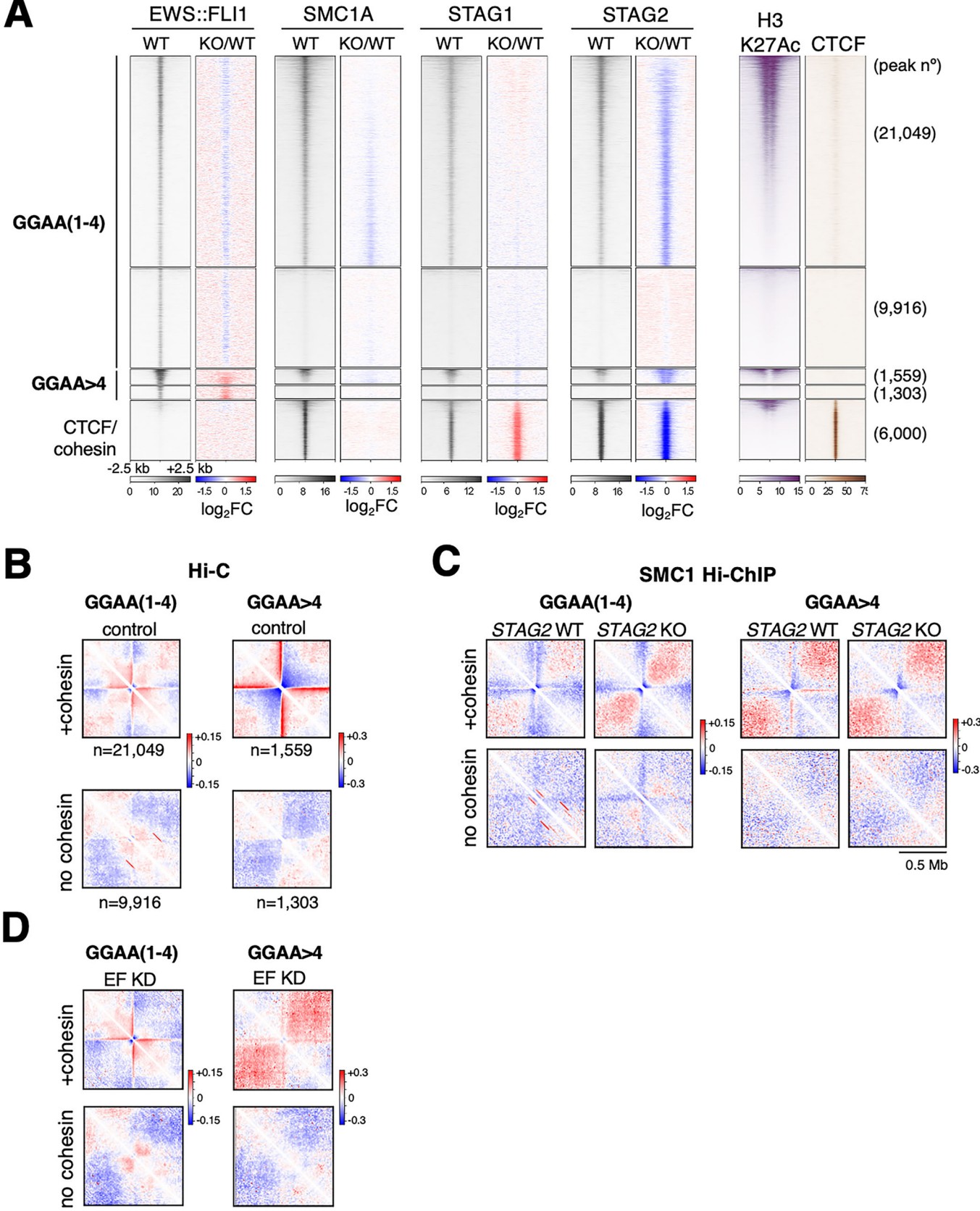

◀ **Figure 3. Cohesin-STAG2 contributes to the establishment of EWS::FLI1 mediated interactions at long GGAA repeats.**

(**A**) Genome-wide distribution of EWS::FLI1 in A673 cells and presence of cohesin, H3K27ac, and CTCF at those sites (gray heatmaps). Changes in this distribution in *STAG2* KO cells are shown for EWS::FLI1 and cohesin (colored heatmaps, log2FC). Sites have been separated according to the number of GGAA repeats (see also Fig. EV3A) and further subdivided according to the presence or absence of cohesin-STAG2. The cluster at the bottom is a fraction of CTCF/cohesin sites provided for comparison. Full map of these sites in Fig. EV3B. (**B–D**) Metaplots that aggregate chromatin interactions from Hi-C (**B**, **D**) or SMC1 Hi-ChIP data (**C**) emanating from GGAA repeats of different length, with or without cohesin, and extending up to 0.5 Mb away in both directions. Numbers (n, below each metaplot) in (**C**) and (**D**) are the same as in (**B**) and come from the analyses shown in (**A**). Color scales represent the ratio of Observed over Expected interactions (log2). (**B**) and (**D**) show interactions in control (**B**) and EWS::FLI1 (EF) KD cells (**D**). For datasets used, see Table EV1.

GGAA repeats (Fig. EV3A). A fraction of EWS::FLI1 peaks was also enriched in cohesin, but to a lesser extent than canonical CTCF/cohesin binding sites and had little CTCF (Fig. 3A). Cohesin-STAG1 occupancy clearly increased at cohesin/CTCF sites, but not at EWS::FLI1 binding sites, in *STAG2* KO cells (Figs. 3A and EV3B). A meta-analysis of high-resolution Hi-C interactions centered at EWS::FLI1 binding sites showed that only GGAA motifs that were also bound by cohesin were connected to distal genomic regions via bidirectional long-range interactions (Fig. 3B, compare cohesin positive and cohesin negative metaplots; Data ref: Sanalkumar et al, 2023). The strength of these interactions correlates directly with the length of the repeats, with the most robust being those emanating from long GGAA repeats. SMC1 Hi-ChIP data revealed a clear decrease in these interactions in *STAG2* KO cells, suggesting that cohesin-STAG1 cannot compensate for the loss of STAG2 and proper contacts with distant regions cannot be established despite increased binding of the oncoprotein to long GGAA repeats (Fig. 3C; Data ref: Adane et al, 2021). The requirement for EWS::FLI1 is specific for contacts emanating from long GGAA repeats, while regions with single or short repeats remain connected and even increase their contacts after depletion of the oncoprotein (Fig. 3D, compare with 3B). Taken together, these results are in agreement with the formation of cohesin STAG2-dependent contacts in response to the presence of EWS::FLI1 in Ewing sarcoma cells, most prominently at long GGAA repeats.

## STAG2 loss impairs chromatin architecture and the associated transcription of EWS::FLI1 target genes

Next, we asked about the nature of these STAG2-mediated interactions and their relevance to transcription. To answer this question, we conducted Promoter Capture Hi-C (PCHi-C), a technique that generates high-resolution genome-wide interaction profiles between promoters and distal regions. We constructed 16 libraries in two independent experiments, each with technical replicates of two A673 cell lines with STAG2 (Parental and WT clone) and two cell lines without STAG2 (KO#1 and KO#2 clones). Over 200,000 statistically robust interactions were obtained, which fulfilled the criteria of having at least five reads and a score >3 (Table EV2). The loop length distribution profiles were very similar for the two biological replicates of each genetic condition, with the STAG2 deficient clones presenting a reduced number of interactions in the size range expected for cohesin-mediated loops (80–800 kb; Fig. 4A). To select the most robust interactions, we restricted subsequent analyses to contacts that were present in both STAG2 proficient cell lines (Parental and WT clone, 54,226 loops) or in both *STAG2* KO clones (37,142 loops). Among these contacts, we considered "common" loops those that were present in three out of

four cell lines while "gained" loops were only present in the *STAG2* KO clones and "lost" loops were only called in STAG2 expressing cells (Fig. 4B, Dataset EV2). Importantly, "common" interactions tended to be weaker in these cells than in STAG2 proficient cells.

To assess the impact of changes in the promoter interactome on gene expression, we focused first on "direct" EWS::FLI1 targets (Dataset EV3). To define these genes, we integrated PCHi-C data, EWS::FLI1 occupancy profiles and gene expression data after EWS::FLI1 knock down (KD) in A673 cells (Table EV1; Data ref: Surdez et al (2021); Data ref: Tomazou et al (2015)). We selected 1429 genes that are deregulated after EWS::FLI1 KD (|log2FC|>0.5) and present EWS::FLI1-bound GGAA motif(s) either at their promoter ("P" in Fig. 4C) or at a distal region connected with the promoter by at least one contact ("distal GGAA repeats" in Fig. 4C). Within the latter, we separated genes with promoters linked to regions with less (short) or more (long) than four GGAA repeats. Genes within the "P" and "distal long GGAA" classes are often downregulated after EWS::FLI1 KD, while those in the "distal short GGAA" class are upregulated (Fig. 4C), consistent with the observed increase in interactions (Fig. 3D). Loss of STAG2 preferentially affected genes with no EWS::FLI1 at the promoter, but interacting with long GGAA repeats (Fig. 4D, left). Most of these interactions were weakened or lost in *STAG2* KO cells, consistent with the results presented in the previous section (Figs. 4E and 3C), and the extent of gene deregulation correlated with the loss of interactions (Fig. 4F). A clear example of this behavior is the *ADRA1D* gene (Fig. 4G; Appendix Fig. S1). The GGAA microsatellite bound by EWS::FLI1 is most likely a site for loading/anchoring cohesin, which, according to the Hi-C map, establishes connections with a downstream CTCF site and the upstream promoter of the *ADRA1D* gene, also bound by CTCF. Only the latter was captured in PCHi-C. In *STAG2* KO cells, however, interactions with the gene promoter are lost and the gene is strongly downregulated (log2FC = −2.43, Dataset EV1). The reduction or loss of chromatin contacts between promoters and distal EWS::FLI1-bound regions with long GGAA repeats can explain the downregulation of six genes within the STAG2-dependent "survival signature" (Fig. 4H). Interactions between promoters and distal short GGAA repeats were less dependent on cohesin-STAG2, and expression of these genes was, on average, less affected by STAG2 loss (Fig. 4D, left, and F). Nevertheless, a group of signature genes belong to this category (Appendix Fig. S2A). We also found that the loss of STAG1 did not affect expression of the EWS::FLI1 target genes (Fig. 4D, right). Finally, 566 genes with EWS::FLI1 at their promoters did not change on average in *STAG2* KO cells, but a few that became significantly downregulated belonged to the "survival signature" (Appendix Fig. S2B).

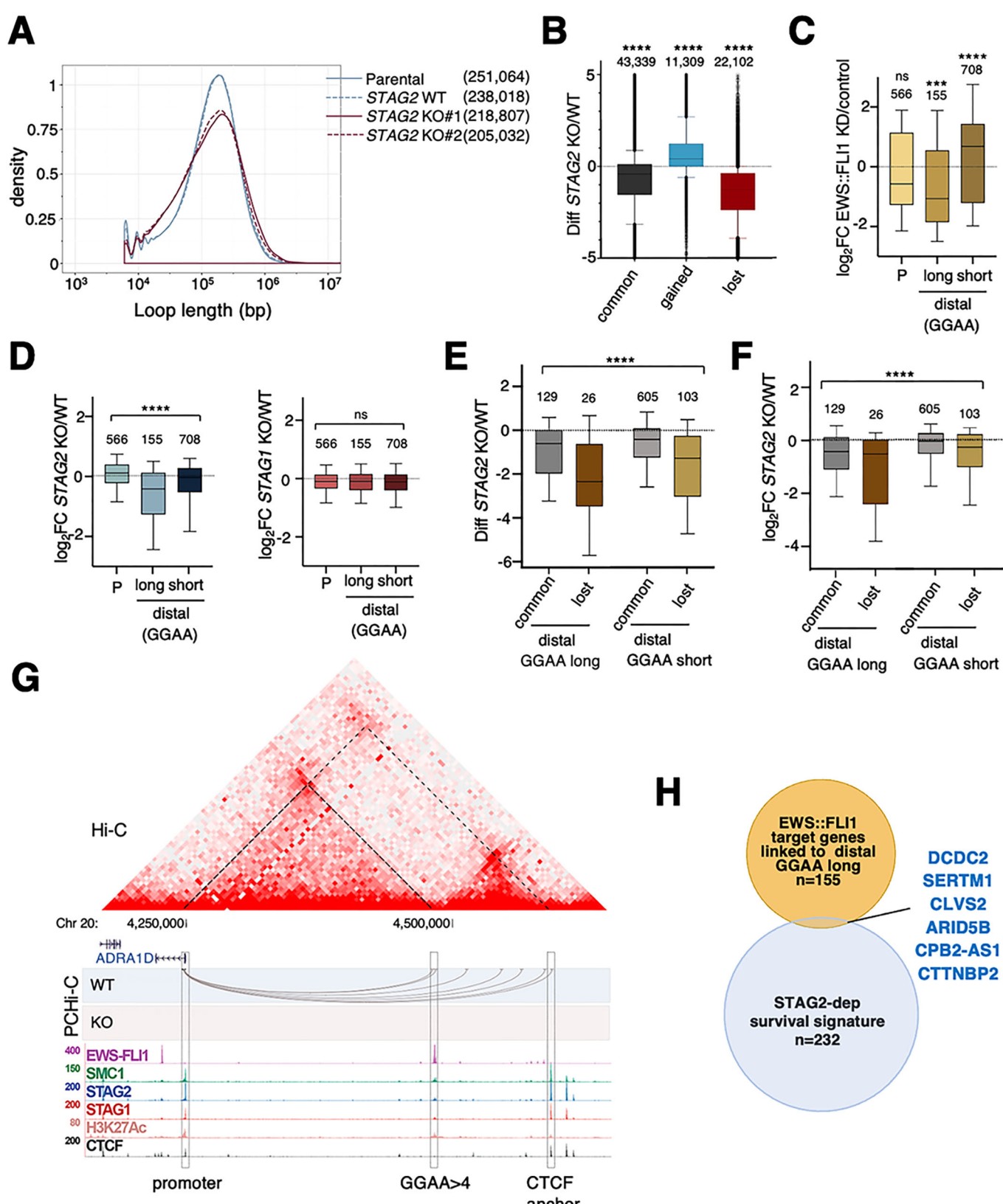

◀ **Figure 4. Cohesin STAG2-mediated looping facilitates contacts between EWS::FLI1-bound long GGAA repeats and target promoters.**

(A) Distribution of loop length according to PCHi-C analyses in the indicated conditions. (B) Changes in contacts between *STAG2* KO and WT conditions in the indicated loops. Mann–Whitney test, ****$p < 0.0001$. Number of contacts in each category is indicated on top of the box. (C) Gene expression changes after EWS::FLI1 KD in genes with EWS::FLI1-bound at GGAA motif(s) present at their promoter (P) and genes whose promoters interact with distal GGAA repeats of the indicated length, also bound by EWS::FLI1. Mann–Whitney test: ns, not significant, $p = 0.2361$; ***$p = 0.0007$; ****$p < 0.0001$. (D) Gene expression changes in the same genes as (C) in *STAG2* KO (left) or *STAG1* KO (right) cells. Kruskal–Wallis test: ns, not significant, $p = 0.5532$; ****$p < 0.0001$. (E) Changes in contacts between promoters and distal GGAA repeats upon STAG2 loss. Kruskal–Wallis test: ****$p < 0.0001$. (F) Correlation between gene expression and contact changes for EWS::FLI1 target genes with no EWS::FLI1 at promoters. Kruskal–Wallis test: ****$p < 0.0001$. In (C–F), number of genes in each category is indicated above the box. For all boxplots in (B–F), the minimum and maximum are typically 1.5 times the Interquartile Range (IQR) from the quartiles, the center value is the median, the box edges are the 25th (Q1) and 75th (Q3) percentiles, and the whiskers extend to the nearest non-outlier points within 1.5 times the IQR. (G) Genomic landscape of the region encompassing *ADRA1D*. From top to bottom, Hi-C matrix, contacts from PCHi-C from the *ADRA1D* promoter and ChIP-seq data. Dotted lines in the Hi-C matrix identify loops emanating from *ADRA1D* promoter. (H) Venn diagram showing overlap between the indicated gene subsets. Source data are available online for this figure.

## Loops established upon STAG2 loss create new EWS::FLI1 target genes

Next, we focused our attention on the genes that are not targets of EWS::FLI1. It has been reported that loss of cohesin STAG2 alters transcription of Polycomb target genes in Ewing sarcoma cells (Adane et al, 2021). This is consistent with our previous data showing that cohesin-STAG2 is important for Polycomb-mediated repression in mouse ES cells (Cuadrado et al, 2019). We also found that gene expression changes in STAG2 deficient cells were inversely correlated with changes in the presence of the H3K27me3 mark in the promoters of both up- and down-regulated genes (Fig. EV4A). To further explore this idea, we generated KO clones for the PRC2 components *SUZ12* and *EZH2* in A673 cells and analyzed their transcriptomes (Fig. EV4B,C). We defined PRC2 target genes as those that were significantly deregulated in both *SUZ12* KO and *EZH2* KO cells in the same direction, which resulted in 1969 upregulated and 726 down-regulated genes (FDR < 0.05; Dataset EV4). A comparison between PRC2 KO and *STAG2* KO deregulated genes confirmed that there was a significant correlation between gene expression changes observed in both conditions (Fig. EV4D). However, the presence or absence of PRC2 is not responsible for the deregulation of some STAG2-dependent genes in which the H3K27me3 mark changes after STAG2 loss. One example is *NR2F1*, one of the genes of the survival signature that is more strongly upregulated in *STAG2* KO cells. Despite visible loss of the repressive mark near the *NR2F1* gene promoter, PRC2 ablation in A673 cells did not significantly change *NR2F1* transcription, suggesting that the loss of Polycomb repression was not the reason for *NR2F1* upregulation in STAG2 deficient cells (Fig. 5A; see also NR2F1 protein levels in EV4B). Instead, we identified two very long interactions (>1 Mb) emanating from the *NR2F1* promoter that are gained in *STAG2* KO cells, one to the left with a FLI1-bound GGAA long repeat labeled by H3K27ac and one to the right with a CTCF site that shows no active or repressive marks (Fig. 5A and Appendix Fig. S3). It is possible that loss of STAG2 weakens the CTCF boundary separating an EWS::FLI1-bound long GGAA repeat and the *NR2F1* promoter in A673 cells (arrow below the Hi-C matrix in Fig. 5A), which, together with the increased processivity of cohesin-STAG1 (see below), would facilitate a contact that upregulates *NR2F1*. Consistent with this hypothesis, EWS::FLI1 KD decreased NR2F1 accumulation in *STAG2* KO cells at both mRNA and protein levels (Fig. 5B). CTCF KD in *STAG2* proficient cells also increased *NR2F1* transcription (Fig. 5C). Among the contacts established from long GGAA repeats bound by EWS::FLI1, those that were lost in *STAG2* KO cells were shorter than those that were maintained, while the size

range was more heterogeneous for the new (gained) contacts and included contacts beyond the 1 Mb TAD range (Fig. 5D). Thus, new long-range contacts between promoters and EWS::FLI1-bound GGAA long repeats may explain some of the transcriptional alterations observed after STAG2 loss.

## STAG2 loss impairs chromatin architecture and associated transcription of EWS::FLI1 independent genes

We next examined changes in gene expression unrelated to EWS::FLI1 in STAG2 deficient Ewing sarcoma cells. Previous studies in other cell types have shown that the downregulation or ablation of STAG2 preferentially affects local chromatin contacts. In agreement with these studies, we observed that interactions between promoters and distal regions mediated by cohesin/CTCF are also differentially affected by STAG2 loss depending on their size, with a reduction in shorter contacts (100–800 kb) and increased contacts between distant (>1 Mb) regions (Fig. 6A). To explore how these changes affect gene expression, we selected 991 STAG2-dependent genes that have E-P contacts anchored by cohesin at both sites and CTCF in at least one of them, and that are not transcriptional targets of EWS::FLI1. For most genes, interactions were still present in *STAG2* KO cells, but were reduced in intensity (Fig. 6B). There was also a fraction of genes that gained and lost contacts. Genes with reduced or lost interactions tended to be downregulated, whereas genes that gained new loops were upregulated in STAG2 deficient cells (Fig. 6C, left). The expression of these genes remained largely unchanged in *STAG1* KO cells (Fig. 6C, right). *RNF141* is one of the genes downregulated in *STAG2* KO cells that showed a clear reduction in contacts between its promoter and distal CTCF sites (Fig. 6D; Appendix Fig. S4). Changes in CTCF/cohesin loops upon STAG2 loss may contribute to the deregulation of up to one-fourth of the STAG2-dependent genes that correlate with poor prognosis in EWS patients, including *RNF141* (Fig. 6E).

## Altered extrusion dynamics in STAG2 deficient Ewing sarcoma cells

Loop extrusion mediated by cohesin requires binding of NIPBL to the complex to activate ATPase activity of the SMC heads (Davidson et al, 2019; Petela et al, 2018). The binding of NIPBL is transient and competes with the binding of PDS5, which pauses loop enlargement (Rhodes et al, 2017; Cuadrado et al, 2022; van Ruiten et al, 2022; Bastié et al, 2022). We investigated how the loss of STAG2 affects the levels of cohesin and its closest regulators in Ewing sarcoma cells, which in turn is likely to affect extrusion dynamics and thereby genome folding. At

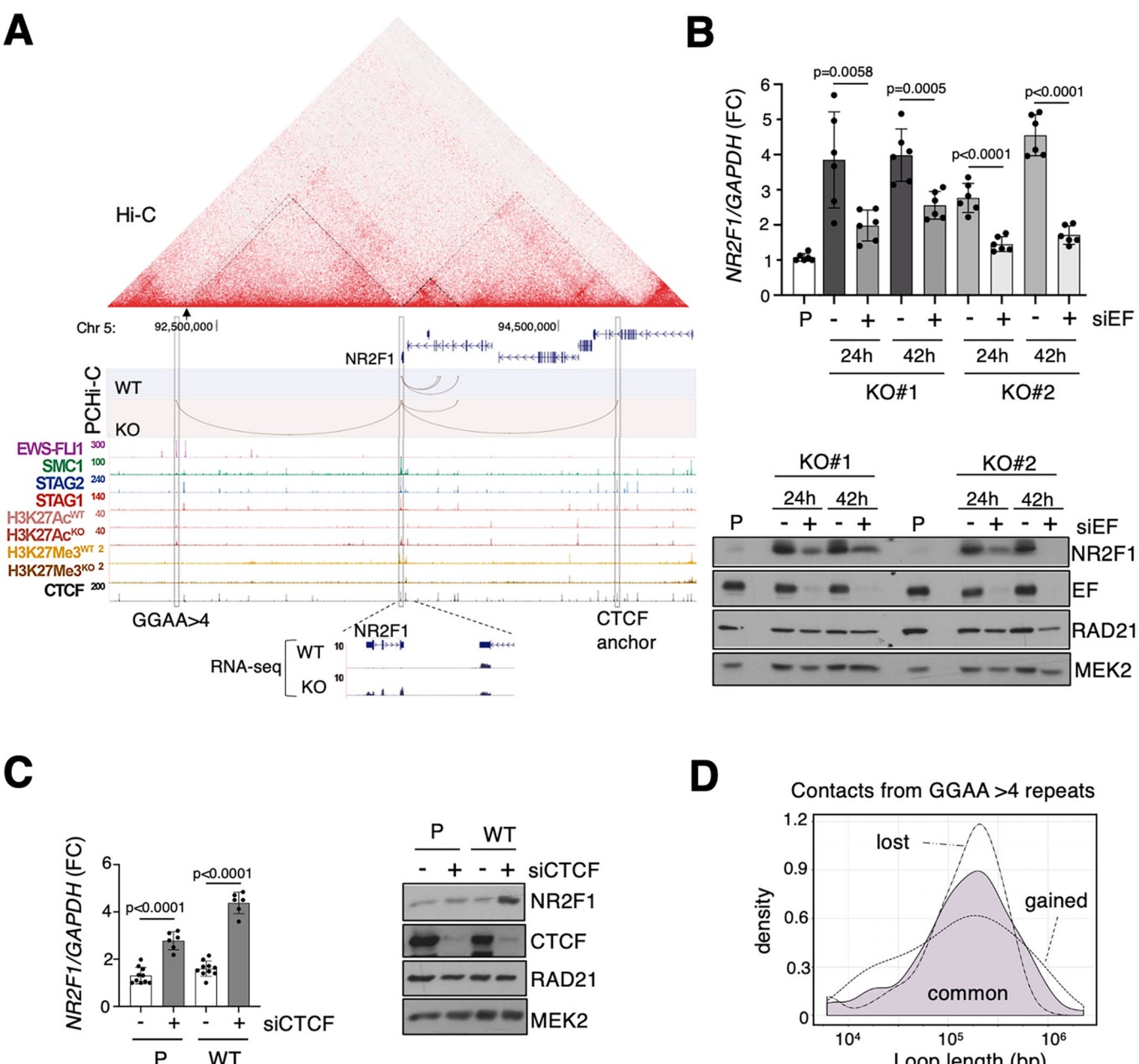

**Figure 5. Loops established upon STAG2 loss create new EWS::FLI1 target genes.**

(A) Genomic landscape of the region encompassing the *NR2F1* gene, an upregulated gene of the survival signature, as in previous Fig. 4G. Reads from RNA-seq for this gene are included at the bottom. (B) Gene expression levels of *NR2F1* measured by qRT-PCR (top) and protein levels assessed by immunoblot analysis of whole-cell extracts (bottom) of *STAG2* KO A673 cells either mock transfected (−) or transfected (+) with siRNA against EWS::FLI1 (siEF). Non-transfected parental A673 cells (P) were used as a reference. MEK2 was used as a loading control. Bar graph represents mean ± SD from n = 6 biological replicates, one-way ANOVA test. (C) As in (B) for Parental (P) and *STAG2* WT A673 cells either mock transfected (−; n = 10) or transfected (+; n = 6) with siRNA against CTCF (siCTCF) for 72 h. Bar graph represents mean ± SD, one-way ANOVA test. (D) Loop length distribution of chromatin contacts anchored at long GGAA repeats that either persist (common, mauve), or are lost or gained in *STAG2* KO A673 cells. Source data are available online for this figure.

the transcriptional level, only *STAG1* and *PDS5B* were significantly upregulated in *STAG2* KO cells (Fig. EV5A). Protein levels were analyzed by immunoblotting of whole-cell extracts (Fig. EV5B), whereas chromatin-bound proteins were compared by flow cytometry (Fig. 7A). *STAG2* KO clones displayed a reduction in RAD21 both in total cell extracts and chromatin-bound protein levels. SMC1 was also

clearly reduced. Thus, STAG1 upregulation cannot compensate for the loss of STAG2, leading to a reduced amount of cohesin on chromatin. A similar decrease on chromatin was observed for WAPL while the reduction in PDS5A was less noticeable. Importantly, NIPBL levels remained unchanged. This means that the ratio of NIPBL:cohesin or NIPBL:PDS5 is higher in *STAG2* KO cells than in *STAG2* WT cells.

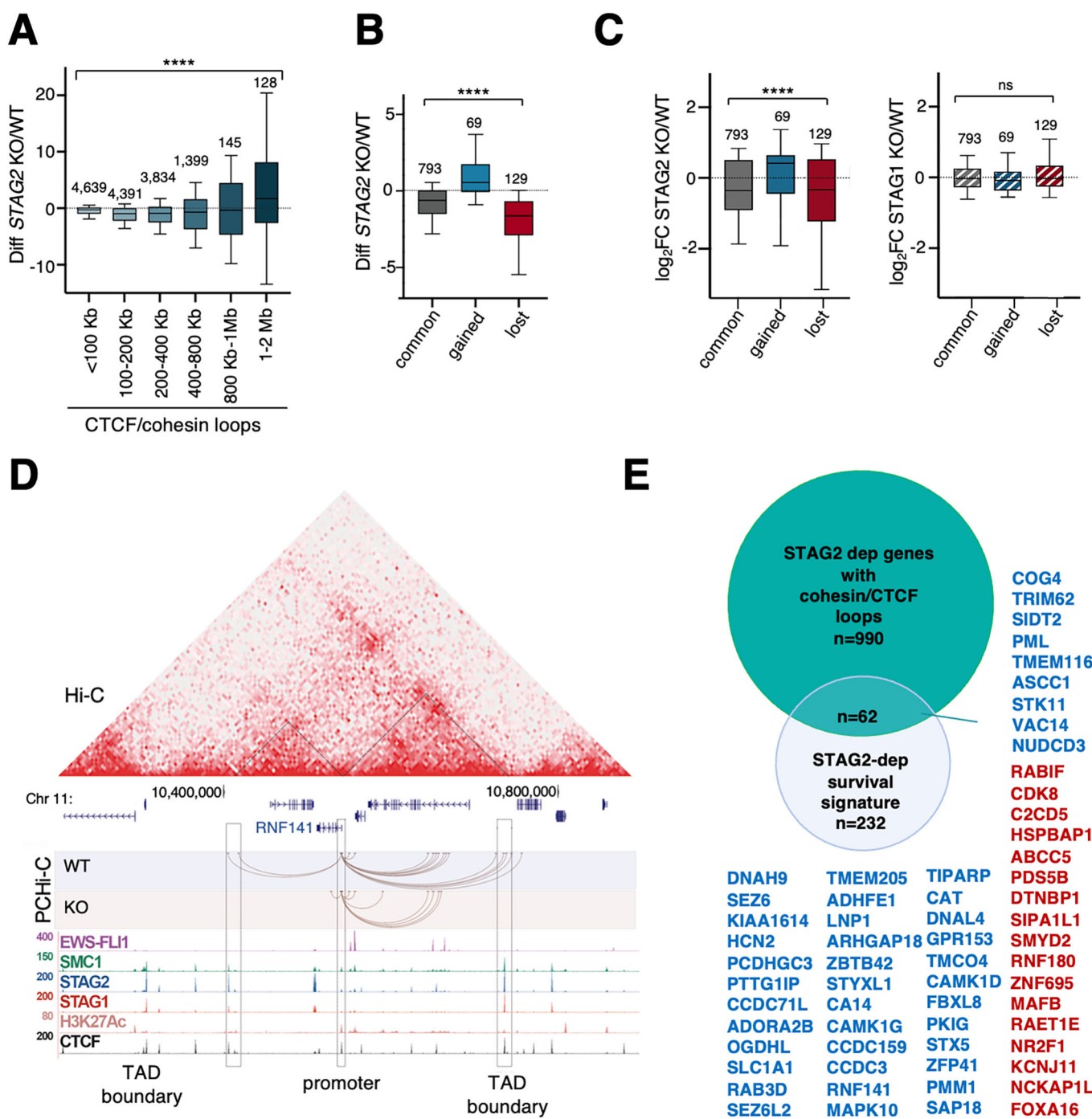

**Figure 6. Gene expression changes independent of EWS::FLI1 that result from changes in CTCF/cohesin loops.**

(A) Changes in all cohesin/CTCF-bordered interactions between promoters and distal regions for different loop sizes upon STAG2 loss. The number of interactions in each category is indicated on the top of the box. Kruskal–Wallis test: ****$p < 0.0001$. (B) Differential interactions between *STAG2* WT and KO cells for cohesin/CTCF loops involving STAG2-dependent genes (number of genes is indicated). Kruskal–Wallis test: ****$p < 0.0001$. (C) Gene expression changes associated with the loops in (B) in *STAG2* KO (left) and *STAG1* KO cells (right). Number of genes in each category is indicated above the box. Kruskal–Wallis test: ns, not significant, $p = 0.2678$; ****$p < 0.0001$. In boxplots in (A–C), the minimum and maximum are typically 1.5 times the Interquartile Range (IQR) from the quartiles, the center value is the median, the box edges are the 25th (Q1) and 75th (Q3) percentiles, and the whiskers extend to the nearest non-outlier points within 1.5 times the IQR. (D) Genomic landscape of the region encompassing *RNF141*, which belongs to the survival signature. (E) Venn diagram showing overlap between the indicated gene subsets. Upregulated genes are in red, downregulated genes are in blue. Source data are available online for this figure.

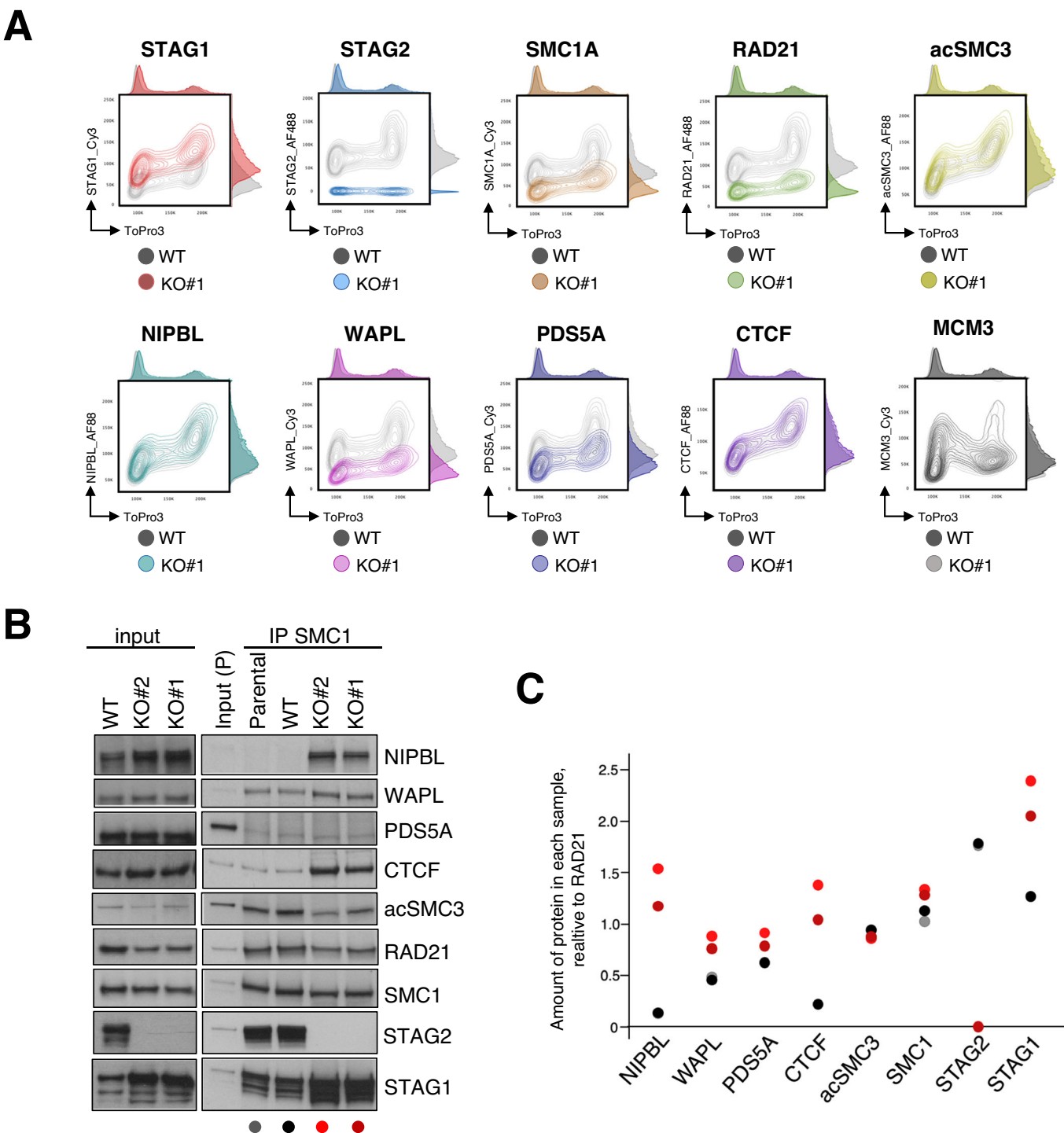

**Figure 7. STAG2 loss alters the balance of cohesin regulators in Ewing sarcoma cells.**

(A) Chromatin-bound levels of cohesin subunits and regulators assessed by Chromoflow flow cytometry. A representative experiment comparing A673 WT and *STAG2* KO#1 is shown. Similar results were obtained comparing WT or Parental cells and *STAG2* KO#1 or KO#2 in two independent experiments. (B) Immunoprecipitation of cohesin with anti-SMC1 from the indicated cell extracts followed by immunoblotting of cohesin and regulators. Colored dots below the panels identify each sample. (C) Quantification of the amount of the indicated proteins co-immunoprecipitated with anti-SMC1, relative to the amount of RAD21, in these four samples. Gray dots often overlap with black dots. Source data are available online for this figure.

Consistent with this, immunoprecipitation of cohesin with an SMC1 antibody pulled down more NIPBL in STAG2 deficient cells than in STAG2 proficient cells (Fig. 7B,C). PDS5A was pulled down to a similarly low extent in all clones while WAPL was more abundant in SMC1 immunoprecipitates from *STAG2* KO cells. We conclude that STAG2 deficient cells have less cohesin on chromatin, but a larger fraction of these complexes is bound to NIPBL at any given time and can perform loop extrusion with increased efficiency. This may explain why there are fewer loops, but also why some longer, probably interTAD loops, are gained in these STAG2 deficient cells.

## Discussion

To understand how the loss of cohesin STAG2 affects Ewing sarcoma patients, we thoroughly compared the transcriptomes of patients and cell lines with and without STAG2. We identified a group of patients that did not carry mutations in *STAG2* but had transcriptomes and survival rates similar to those of patients with *STAG2* mutations. Some of these cases show reduced *STAG2* expression, perhaps as a result of mutations in cis-regulatory elements that drive *STAG2* transcription or promoter methylation. A recent analysis of a different patient cohort identified several cases in which the loss of STAG2 protein, observed by immunohistochemistry in patient samples, occurred in the absence of *STAG2* mutations (Shulman et al, 2022). Consistent with our data, these patients had adverse prognoses. Thus, loss of STAG2 staining in patient biopsies of primary tumors is a better predictor of outcome than finding mutations or reduced *STAG2* expression.

Previous reports have proposed that loss of STAG2 attenuates the EWS::FLI1 oncogenic program and results in an EWS-FLI "low state" that enhances cell migration and metastatic behavior (Surdez et al, 2021; Adane et al, 2021; Franzetti et al, 2017). Indeed, several STAG2-dependent genes are EWS::FLI1 targets that are downregulated upon STAG2 loss. This is not a consequence of decreased binding of the oncoprotein to GGAA repeats. Instead, we show that EWS::FLI1 occupancy increases at long GGAA repeats, presumably acting as enhancers, but its ability to reach distal target genes is severely compromised in the absence of cohesin-STAG2 mediated looping.

Altered EWS::FLI1 dependent transcription is unlikely to explain all the changes that are important in promoting the more aggressive phenotype of Ewing sarcoma in patients lacking STAG2. Only one-fourth of the genes within the STAG2-dependent survival signature that we defined are EWS::FLI1 targets. Cohesin/CTCF-anchored interactions between promoters and distal regulatory elements are also modified upon STAG2 loss, which affects gene expression beyond the EWS::FLI1 target genes. CTCF-HiChIP data showed that A673 cells lacking STAG2 present reduced stripes but enhanced corner peaks (Surdez et al, 2021). We observed that cohesin complexes on chromatin are significantly reduced in *STAG2* KO cells, as the increase

of STAG1 is not sufficient to compensate for the lack of STAG2. The levels of some cohesin regulatory factors such as WAPL and PDS5 are concomitantly decreased on chromatin while NIPBL amounts are similar in STAG2 proficient and deficient cells. Less cohesin on chromatin and a higher NIPBL:cohesin ratio likely reduce the probability of cohesin-cohesin collisions and increase cohesin processivity, thus leading to fewer but longer loops (van Ruiten et al, 2022). This, together with the prolonged retention of cohesin-STAG1 at CTCF sites that constitute the borders of corner peaks, a distinctive feature of STAG1 complexes, could explain the CTCF-HiChIP data described above (Surdez et al, 2021; Wutz et al, 2020). Nevertheless, loop extension beyond a domain boundary may facilitate contacts that are normally restricted in *STAG2* WT cells. This appears to be the case with EWS::FLI1 driven upregulation of *NR2F1* in *STAG2* KO cells.

In addition to providing mechanistic understanding of the consequences of STAG2 loss for the transcriptome of Ewing sarcoma cells, our results identified a gene signature that could be further refined to select biomarkers useful for predicting disease progression and/or serve as therapeutic targets. We did not find *CD274*, the gene encoding PD-L1, among the STAG2-dependent genes in A673 cells. This is in contrast to the recognition of cohesin and CTCF as potent suppressors of PD-L1 expression in mammary epithelial and myeloid cells (Oreskovic et al, 2022). The development of in vivo models and analysis of the immune environment in patients with and without *STAG2* mutations will be important future directions for exploring the contribution of immune evasion to aggressive Ewing sarcoma and its implications for immunotherapy (Visser et al, 2023). We cannot exclude the possibility that loss of STAG2 affects Ewing sarcoma cells through mechanisms beyond gene expression regulation, including those related to genome instability. Previous studies have suggested that Ewing sarcoma cells exhibit increased replication stress and R-loop accumulation, which may be exacerbated in the absence of cohesin-STAG2 (Mondal et al, 2019; Nieto-Soler et al, 2016; Gorthi et al, 2018; Martin et al, 2022; preprint: Olmedo-Pelayo et al, 2023). In summary, loss of STAG2 may contribute to the worse outcome of Ewing sarcoma patients through multiple mechanisms that remain to be elucidated in order to provide novel treatment options for cohesin-mutant patients.

## Methods

### Cell lines

Ewing sarcoma cell lines A673, A4573, and SK-N-MC were obtained from Dr. Enrique de Álava. Cell identity was verified by short tandem repeat profiling with the following reference samples A673: CVCL 0080 (100%); A4573: CVCL_7150 (96.8%), SK-NM-C:

**Reagents and tools table**

| Reagent/Resource | Reference or Source | Identifier or Catalog Number |
|---|---|---|
| Experimental Models | | |
| A673 | ATCC/E. de Álava (IBIS) | cat#CRL-1598 |
| A4573 | E. de Álava (IBIS) | – |
| SK-NM-C | ATCC/E. de Álava (IBIS) | cat#HTB-10 |

| Reagent/Resource | Reference or Source | Identifier or Catalog Number |
|---|---|---|
| **Recombinant DNA** | | |
| pSpCas9(BB)-2A-Puro | Addgene/Raúl Torres (CNIO) | cat#62988 |
| **Antibodies** | | |
| Rat monoclonal anti-STAG1 IB: 1:10 diluted supernatant | Custom made, Kojic et al (2018), clone SUSI 63B | |
| Rabbit polyclonal anti-STAG1 FC: 4 µg/ml; IB: 2 µg/ml | Custom made, Remeseiro et al (2012a) | |
| Mouse monoclonal anti-STAG2 FC: 2 µg/ml; IB: 0.4 µg/ml | SCBT | cat#sc-81852 |
| Rabbit polyclonal anti-STAG2 (C-term) IB: 1 µg/ml | Custom made, Remeseiro et al (2012a) | |
| Rabbit polyclonal anti-STAG2 (N-term) IB: 2 µg/ml | Custom made, this study | |
| Rabbit polyclonal anti-SMC1A FC: 4 µg/ml; WB: 1 µg/ml | Custom made, Remeseiro et al (2012b) | |
| Rabbit polyclonal anti-RAD21 IB: 2 µg/ml | Custom made, Carretero et al (2013) | |
| Mouse monoclonal anti-RAD21 clone 53A303; FC: 2 µg/ml | Merck | cat#05-908 |
| Mouse monoclonal anti-NIPBL FC: 2 µg/ml | SCBT | cat#sc-374625 |
| Rabbit polyclonal anti-NIPBL IB: 5 µg/ml | Custom made, Alonso-Gil et al (2023) | |
| Rabbit anti-MAU2/SCC4 IB: 1:1000 | Abcam | cat#ab183033 |
| Mouse monoclonal anti-CTCF FC: 2 µg/ml | SCBT | cat#sc-271474 |
| Rat monoclonal anti-CTCF IB: Undiluted supernatant | Custom made, clone MARS159A/D, Alonso-Gil et al (2023) | |
| Rabbit polyclonal anti-PDS5A IB: 2 µg/ml, FC:2 µg/ml | Custom made, Carretero et al (2013) | |
| Rabbit polyclonal anti-PDS5B IB: 5 µg/ml | Custom made, Morales et al (2020) | |
| Mouse monoclonal anti-acetyl SMC3 IB: 1:500; FC: 1:50 | Custom made, Nishiyama et al (2010) | |
| Mouse monoclonal anti-ESCO1 IB: 1:200 | Custom made, Minamino et al (2015) | |
| Rat monoclonal anti-WAPL IB: 1:10 supernatant | Custom made, clone WAPI 432E, Morales et al (2020) | |
| Rabbit polyclonal anti-WAPL FC: 4 µg/ml | Custom made, Kojic et al (2018) | |
| Rabbit anti-HDAC8 IB: 1:500 | Abcam | cat#ab137474 |
| Rabbit anti-FLI1 IB: 1:300 | Abcam | cat#ab15289 |
| Rabbit anti-NR2F1 IB: 1:1000 | Abcam | cat#ab181137 |
| Rabbit anti-Suz12 IB: 1:1000 | Bethyl | cat#A302-407A |
| Mouse monoclonal anti-EZH2 (AC22) IB: 1:1000 | CST | cat#3147 |
| Rabbit anti-H3K27me3 C36B11 ChIP: 5 µg in 600 µg chromatin | CST | cat#9733 |
| Mouse monoclonal anti-MEK2 WB: 1:1000 | BD Biosciences | cat#AB_397631 |
| **Oligonucleotides and other sequence-based reagents** | | |
| siRNA EWS::FLI1 GGCAGCAGAACCCUUCUUdGdC | Dharmacon | |
| hCTCF ON-TARGETplus SMARTpool | Dharmacon | L-020165-00 |
| hSTAG2 ON-TARGETplus SMARTpool | Dharmacon | L-021351-00 |

| Reagent/Resource | Reference or Source | Identifier or Catalog Number |
|---|---|---|
| DNA primer for repair 5′G*AGTTGCTCAGCTTTATTTTGGATCATGTCTTCATTGAAC AGGATGATGATAATAATAGTGCAGGTAATTTTATTGCCATC TTTTTATTAAATCTGTGT*C*C 3′ | Sigma | |
| PCR primer NR2F1 fw1: 5′TGGCCTTCATGGACCACATC3′ | IDT Technologies | |
| PCR primer NR2F1 rev1: 5′GACTTCTCCTGCAGGCTCTC3′ | IDT Technologies | |
| PCR primer NR2F1 fw2: 5′AGCAGGTGGAGAAGCTCAAG3′ | IDT Technologies | |
| PCR primer NR2F1 rev2: 5′CTCACGTACTCCTCCAGTGC3′ | IDT Technologies | |
| PCR primer GAPDH fw: 5′TGCACCACCAACTGCTTAGC3′ | IDT Technologies | |
| PCR primer GAPDH rev: 5′GAGGGGCCATCCACAGTCTTC3′ | IDT Technologies | |
| gRNA STAG1: 5′AGATCGATTCAATCATTCTG3′ | IDT Technologies | |
| gRNA STAG2: 5′AATTTCGACATACAAGCACCC3′ | IDT Technologies | |
| gRNA STAG2_A4573: 5′ATTTCGACATACAAGCACCC3′ | IDT Technologies | |
| gRNA SUZ12-1: 5′AGAGAAAATGTTTCGAATGG3′ | IDT Technologies | |
| gRNA SUZ12-2: 5′CTATAGATTTCTTCGAACT3′ | IDT Technologies | |
| gRNA EZH2-1: 5′GTCATAGTAAGTGCCAATG3′ | IDT Technologies | |
| gRNA EZH2-1: 5′GGTAACACTGTGGTCCACA3′ | IDT Technologies | |
| **Chemicals, Enzymes and other reagents** | | |
| DharmaFECT reagent 1 | Dharmacon/Horizon | T-2001-03 |
| Gibco Opti-MEM I Reduced Serum Media | ThermoFisher | cat#31985047 |
| SuperScript™ II Reverse Transcriptase | ThermoFisher | cat# 8064014 |
| RNeasy Mini Kit | Qiagen | cat#74104 |
| SYBR Green PCR Master Mix | ThermoFisher | cat# 4309155 |
| **Software** | | |
| FlowJo v10 | FLOWJO | |
| BD FACSDiva | BD Biosciences | |
| 'Bowtie2' (version 2.4.2) | Langmead and Salzberg, 2012 | |
| GATK4 (version 4.1.9.0) | https://gatk.broadinstitute.org/hc/en-us | |
| MACS2 (version 2.2.7.1) | Zhang et al, 2008 | |
| Coolpup.py | Flyamer et al, 2020 | |
| ChiCAGO | Cairns et al, 2016 | |
| FastQC | http://www.bioinformatics.babraham.ac.uk/projects/fastqc/ | |
| Nextpresso | Graña et al, 2018 | |
| GenePattern | Chapman et al, 2006 | |
| **Other** | | |
| ABI Prism® 7900HT | Applied Biosystems | |
| BD LSRII Fortessa | BD Biosciences | |
| Covaris S220 | Covaris | |
| Illumina NextSeq 500 | Illumina | |
| Arima HiC kit+ Human Promoter Capture HiC Kit | Arima Genomics | cat#A510008 |
| NZY Total RNA Isolation kit | NZYtech | MB13402 |
| NEBNext Single Cell/Low Input RNA Library Prep" kit | New England Biolabs | cat#E6420 |
| NEBNext Ultra II FS DNA Library Prep Kit for Illumina" | New England Biolabs | cat#E7805 |
| NuPAGE™ 3–8% Tris-Acetate gels | ThermoFisher | cat#EA0375PK2 |

CVCL_0530 (100%). A673 and A4573 cells were cultured at 37 °C under 90% humidity and 5% $CO_2$ in DMEM supplemented with 10% FBS and 1% penicillin-streptomycin. SK-N-MC cells were cultured in EMEM supplemented with 10% FBS and 1% penicillin-streptomycin under the same conditions.

## Antibodies

Antibodies for immunoblotting, immunoprecipitation, ChIP-seq and flow cytometry are listed in "Reagents and tools" Table. One antibody was raised in rabbits against a recombinant fragment

containing the initial 220 amino acids of mouse STAG2 protein. This and all the other custom-made rabbit polyclonal antibodies were affinity purified before use.

## CRISPR-Cas9 editing and siRNA

A673 cells expressing dox-inducible Cas9 (A673_iCas9) from the *AAVS1* locus (clone WT) were used to generate *STAG1, STAG2, SUZ12,* and *EZH2* KO clones as described (Alonso-Gil et al, 2023). For *STAG2* KO clones, genomic DNA was amplified and sequenced to identify the mutations generated. To restore STAG2 expression in A4573 cells, cells were co-transfected with 2 µg of single-stranded phosphorylated oligonucleotide template (STA-G2_A4573r) and 6 µg of pSpCas9(BB)-2A-Puro expressing Cas9 and puromycin resistance, into which gRNAs for STAG2 editing were cloned (see "Reagents and tools" Table). Forty-eight hours after transfection, cells were selected with 1 µg/ml puromycin for 72 h, and cellular clones were isolated and analyzed for STAG2 expression by immunoblot. For knock down experiments, cells were transfected with 10–50 nM siRNAs using DharmaFECT reagent 1 and Gibco Opti-MEM I Reduced Serum Media. Cells were harvested 24, 42, or 72 h after transfection (depending on the experiment) and analyzed by immunoblotting and/or quantitative RT-PCR.

## Whole-cell extract preparation and immunoblotting

Whole-cell extracts for immunoblot analyses were prepared in RIPA buffer. Alternatively, cells were resuspended in SDS–PAGE loading buffer at $10^7$ cells/ml, sonicated and boiled before fractionation by SDS–PAGE. NuPAGE™ 3–8% Tris-Acetate gels were used in some cases. Gels were transferred to nitrocellulose membranes in Transfer buffer I (50 mM Tris, 380 nM Glycine, 0.1% SDS, and 20% methanol) for 1 h at 100 V and analyzed by immunoblotting.

## Immunoprecipitation

For cohesin immunoprecipitation, around 15 million cells were lysed on ice for 30 min in Lysis buffer [0.5% NP-40 in TBS supplemented with 0.5 mM DTT, 0.1 mM PMSF and 1X complete protease inhibitor cocktail] followed by sonication and benzonase digestion (1U per µl of extract) for 30 min at 4 °C. Next, NaCl was added to 0.3 M and the extract rotated for additional 30 min at 4 °C. Salt concentration was lowered to 0.1 M NaCl by dilution and glycerol added to 10% final concentration. Extracts were incubated with SMC1 antibody crosslinked to agarose A magnetic beads with dimethyl pimelimidate (30 µl beads bound to 15 µg of SMC1 antibody for 15 million cells) and rotated overnight at 4 °C. The beads were washed 6 times with 20 vol of lysis buffer, at least two times with lysis buffer containing 0.5% NP40, eluted in SDS-DTT gel loading buffer for 5 min at 95 °C, and then analyzed by immunoblotting.

## Chromoflow flow cytometry

To analyze chromatin bound proteins, "Chromoflow" flow cytometry was performed as described (Alonso-Gil and Losada, 2023). In brief, cells were treated for 5 min with a low salt

extraction buffer (0.1% Igepal CA-630, 10 mM NaCl, 5 mM MgCl₂, 0.1 mM PMSF, 10 mM Potassium Phosphate buffer pH 7.4) and fixed in 1% PFA final concentration. Four different samples (P, WT, KO#1 and KO#2) were stained with increasing dilutions of Pacific Blue, pooled, blocked and consecutively incubated with primary and secondary antibodies. DNA was stained overnight with 125 nM ToPRO3-iodide 642/661 in PBS. Cells were analyzed on a BD LSRII Fortessa flow cytometer using BD FACSDiva software and four different lasers: *680/30_R* laser for ToPRO3 (DNA), *450/50_V* for Pacific Blue (barcoding), *586/15_YG* for Cy3-labeled secondary antibody and *525/50_B* laser for Alexa fluor 488-labeled secondary antibody. For statistical analysis, single-cell cycles were gated and at least 10,000 cells were recorded for each population in a barcoded sample. For imaging data, the same number of events was exported for each barcoded population using the FlowJo v10 software. Data quality and fluorescence compensation were assessed to correct for overlap in the emission spectra. Finally, the conditions were merged to compare the behavior of the protein of interest throughout the cell cycle.

## Quantitative RT-PCR

cDNAs were generated using Superscript II Reverse Transcriptase from total RNA (RNeasy Mini Kit) and qRT-PCR analyses were performed using the SYBR Green PCR Master Mix and an ABI Prism® 7900HT instrument. The reactions were performed in triplicate for each sample and samples were obtained from at least three experiments. The expression was normalized to GAPDH using the ΔΔCt method. Two primer pairs were used to assess *NR2F1* expression.

## Chromatin-immunoprecipitation (ChIP)-seq

A673 cells grown to high confluence were crosslinked with 1% formaldehyde for 15 min at room temperature. After quenching the reaction with 0.125 M Glycine, the fixed cells were washed twice with PBS containing 1 µM PMSF and protease inhibitors. Cells were lysed in lysis buffer (1% SDS, 10 mM EDTA, 50 mM Tris-HCl pH 8.1) at a concentration of $2 \times 10^7$ cells/ml. Sonication was performed using a Covaris S220 (shearing time 20 min, 20% duty cycle, intensity 6, 200 cycles per burst and 30 s per cycle) in a minimum volume of 2 ml. Chromatin from $10^7$ cells was incubated with 5 µg of the antibody as described (Kojic et al, 2018). For calibration, 5% of chromatin from mouse ES cells was added to the human chromatin. For library preparation, at least 5 ng of DNA were processed through subsequent enzymatic treatments using "NEBNext Ultra II FS DNA Library Prep Kit for Illumina". Briefly, a short fragmentation of 10 min was followed by end-repair, dA-tailing, and ligation to adapters. The adapter-ligated libraries were completed using limited-cycle PCR (8–12 cycles). The resulting average fragment size was 300 bp, of which 120 bp corresponded to the adapter sequences. Libraries were applied to an Illumina flow cell for cluster generation and sequenced on an Illumina NextSeq 500 (with v2.5 reagent kits) following manufacturer's recommendations.

## ChIP-sequencing analysis

Alignment of reads to the reference human genome (hg38) was performed using 'Bowtie2' (version 2.4.2) under default settings

(Langmead and Salzberg, 2012). Duplicates were removed using GATK4 (version 4.1.9.0) and peak calling was carried out using MACS2 (version 2.2.7.1) after setting the $q$ value (FDR) to 0.05, using the '–extsize' argument with the values obtained in the 'macs2 predictd' step (Zhang et al, 2008). For analysis of calibrated ChIP-seq, profiles for each antibody were normalized by coverage and then multiplied by the occupancy ratio $(OR) = (W_m IP_h)/(W_h IP_m)$, where $W_h$ and $IP_h$ are the number of reads mapped to the human genome from input (W) and immunoprecipitated (IP) fractions, and $W_m$ and $IP_m$ are reads mapped to the mouse genome from the input and IP fractions used for calibration. Calibration was performed only for the new data generated in this study. For Fig. EV3A, EWS::FLI1 ChIP-Seq data in A673 WT and STAG2 KO conditions from GSE116495 and GSE133228 were analyzed. After peak calling, 33,967 unique peaks that did not intersect with the hg38 blacklist and with $q$ value $\geq 5$ were selected, and the number of GGAA repeats within each peak was calculated, allowing a maximum distance of 5 nucleotides between GGAA motifs.

## Analyses of Hi-C and Hi-ChIP data

Coolpup.py was used to generate metaplots centered at the EWS::FLI1 sites using the available Hi-C and Hi-ChIP data listed in Table EV1 (Flyamer et al, 2020). A padding window of 500-kb and the " –local" parameter were used in all metaplots.

## Promoter Capture Hi-C

Promoter Capture Hi-C was performed using the Arima HiC Kit (A510008) and the Arima Human Promoter Capture HiC Kit, which contains probes for 23,711 human promoters (GRCh38 ensemble database, version 95). We used 7–8 million cells for each condition (P, WT, KO#1 and KO#2) and 10–11 PCR cycles to prepare the Hi-C libraries. Post-capture libraries were amplified using 11–12 PCR cycles.

## Promoter Capture Hi-C analysis

Two independent experiments were performed with two technical replicates for each of the four cell lines. Each sample was analyzed using the Arima Capture Hi-C pipeline and aligned to the hg38 genome at 3-kb resolution. The resulting BAM files from the technical replicates and different experiments were merged to create a single file for each cell line (four in total). To allow comparisons between the four conditions, we performed down-sampling to equalize the number of reads aligned with the Arima probes (% on-target). Loops were called using ChiCAGO for the individual cell lines with at least five reads in the interaction and a score >3, resulting in more than 200,000 loops called in each condition (Cairns et al, 2016). Of these, 54,226 loops were detected in the P and WT conditions, whereas 37,142 were detected in STAG2 null clones. To classify these loops, we established the following criteria: if a loop was called in 3 out of 4 (STAG2 WT and KO), it was classified as "common", while loops called only in the STAG2 KO condition were classified as "gained" and those present only in the STAG2 WT condition were classified as "lost". To calculate loop strength, we first merged the BAM files from Parental and WT, and those from the two STAG2 KO clones, and converted them into "mcool" files for easier handling. These matrices were

row-normalized by dividing each row by the total sum of the rows, and then normalized by decay by distance, using only those rows that contained a bait.

## Bulk RNA sequencing

Total RNA was extracted using the NZY Total RNA Isolation kit following the manufacturer's instructions. Total RNA samples (500 ng) were processed using the "NEBNext Single Cell/Low Input RNA Library Prep" kit. RNA Quality scores were 9.9 on average (range 9.1–10) when assayed on a PerkinElmer LabChip analyzer. Briefly, oligo(dT) primed reverse transcription with a template switching reaction was followed by double-stranded cDNA production using limited-cycle PCR. Non-directional sequencing libraries were completed with the "NEBNext Ultra II FS DNA Library Prep Kit for Illumina" and subsequently analyzed on an Illumina NextSeq 550 with v2.5 reagent kits following the manufacturer's protocols.

## RNA-sequencing analysis

Fastq files with 86-nt single-end sequenced reads were quality-checked with FastQC (S. Andrews, http://www.bioinformatics.babraham.ac.uk/projects/fastqc/) and aligned to the human genome (hg38) with Nextpresso executing TopHat-2.0.0, Bowtie 0.12.7 and Samtools 0.1.16 allowing two mismatches and five multi-hits (Graña et al, 2018). Finally, differentially expressed genes were filtered using an FDR < 0.05.

## Analysis of patient cohorts

Data from three publicly available patient cohorts were used in this study (Tirode et al, 2014; Savola et al, 2011; Volchenboum et al, 2015). For single-sample Gene Set Enrichment Analysis (ssGSEA; Fig. 1D), the online tool GenePattern was used (Chapman et al, 2006) with transcriptomic data from the ICG dataset (Data ref: Tirode et al, 2014) and CGP gene sets. Differentially expressed gene sets were obtained by comparing STAG2 MUT with STAG2 WT, excluding STAG2 WT* cases. To classify patients in "signature-like" and signature-different" expression, we used the 232 genes that were differentially expressed in patients and in A673 clones. Initially, a "STAG2 KO model" was constructed, wherein each upregulated gene had the maximum value in the dataset for that gene, while each downregulated gene had the minimum value. This model represents an extreme phenotype associated with STAG2 loss. Conversely, a "STAG2 WT model" was established using the opposite rationale: for each upregulated gene, the model had the minimum value, and vice versa for downregulated genes. Subsequently, unsupervised clustering (k-means) with two groups and centroids initialized to STAG2 WT and STAG2 KO model values was employed to categorize patients of two patient cohorts (Data ref: Savola et al, 2011; Data ref: Volchenboum et al, 2015) as either "signature-like" or "signature-different". Finally, PCA segregated patients according to the previously determined clusters (Fig. 2B). Overall survival probabilities were calculated for patients in each group. This unbiased analysis was performed independently in the two cohorts and after merging the two datasets using the 'sva' R package to mitigate biases stemming from different datasets (Fig. EV2 and Fig. 2C, respectively).

# Data availability

The ChIP-seq, RNA-seq, and PCHi-C data generated in this study have been deposited in GEO (GSE267223). Additional datasets used are listed in Table EV1.

The source data of this paper are collected in the following database record: biostudies:S-SCDT-10_1038-S44319-024-00303-6.

# Peer review information

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

## Acknowledgements

We thank O. Martínez Tirado for critically reading the manuscript and members of the DNA Replication group at CNIO for discussion. We thank E. de Álava for cell lines, K. Shirahige for antibodies for ESCO1 and acetyl SMC3 and R. Torres for plasmid pSpCas9(BB)-2A-Puro. We acknowledge the contribution of the Genomics and Flow Cytometry Units at CNIO. This research has been possible thanks to the generous support of Fundación Científica AECC (PROYE20046LOSA) and to grants PID2019-106499RB-I00 and PID2022-139333NB-I00 funded by MCIN/AEI/10.13039/501100011033 and by the ERDF A way to making Europe, by the European Union. MJA has been additionally supported by a postdoctoral fellowship from CNIO Friends. ISA has benefited from a predoctoral contract from Comunidad de Madrid (PEJ-2020-AI/BMD-18406) and is currently supported by the FPI-Severo Ochoa Program (PRE2021-100005). MS is also the recipient of a predoctoral FPI contract (PRE2020-091957).

## Author contributions

**Daniel Giménez-Llorente**: Data curation; Software; Formal analysis; Investigation; Methodology; Writing—review and editing; Computational analyses. **Ana Cuadrado**: Conceptualization; Supervision; Investigation; Methodology; Writing—original draft; Writing—review and editing. **Maria José Andreu**: Formal analysis; Investigation; Writing—review and editing. **Inmaculada Sanclemente-Alamán**: Formal analysis; Investigation; Writing—review and editing. **Maria Solé-Ferran**: Investigation; Methodology; Writing—review and editing. **Miriam Rodríguez-Corsino**: Investigation; Methodology; Writing—review and editing. **Ana Losada**: Conceptualization; Supervision; Funding acquisition; Visualization; Writing—original draft.

Source data underlying figure panels in this paper may have individual authorship assigned. Where available, figure panel/source data authorship is listed in the following database record: biostudies:S-SCDT-10_1038-S44319-024-00303-6.

## Disclosure and competing interests statement

The authors declare no competing interests.

# Expanded View Figures

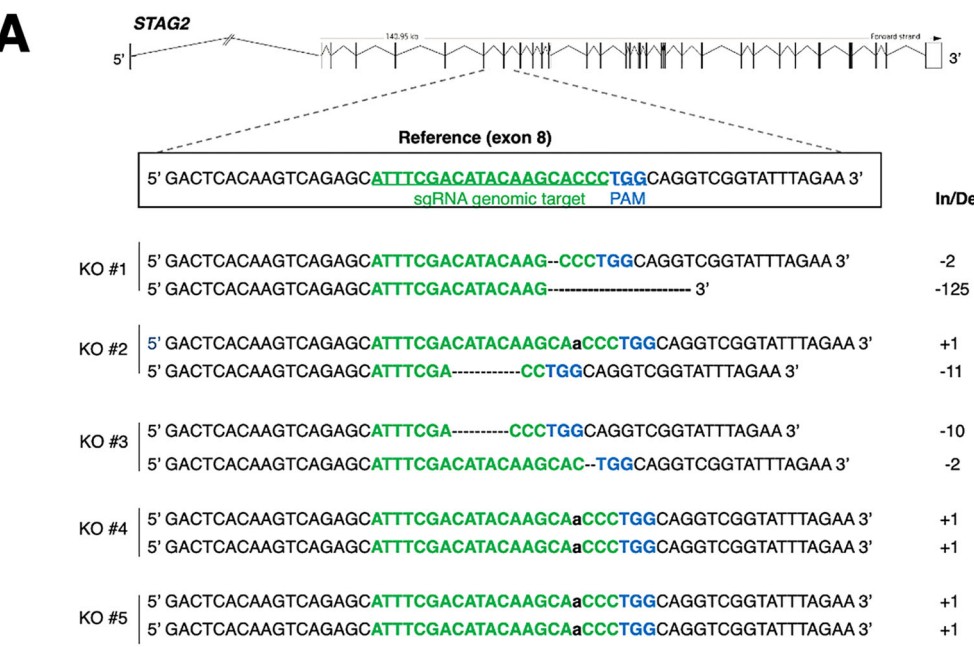

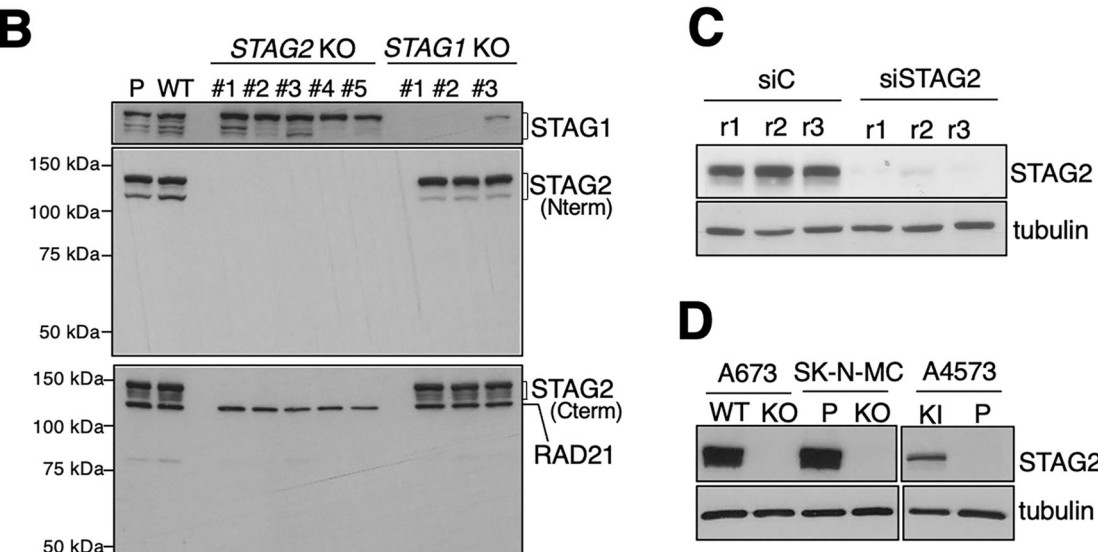

**Figure EV1.  Generation and characterization of Ewing sarcoma cell lines with and without STAG2.**

(A) Mutations generated by CRISPR/Cas9 editing in *STAG2* gene. All clones were generated from a clone carrying an inducible Cas9 at the *AAVS1* locus (*STAG2* WT). (B) Immunoblot analysis of whole-cell extracts of parental (P) A673 cells and clones prepared in RIPA buffer. (C) A673 cells were mock transfected (control, siC) or transfected with siRNA against STAG2 (siSTAG2) and used for RNA-seq (3 replicates per condition, r1 to r3). Tubulin is used as loading control. (D) Immunoblot analysis of whole-cell extracts of Ewing sarcoma cells SK-N-MC and A4573 with and without STAG2, used for RNA-seq. A673 WT and *STAG2* KO#2 were used for comparison. Tubulin as loading control. Parental (P) SK-N-MC cells express STAG2 while A4573 cells do not, and they were edited to generate *STAG2* KO and KI clones, respectively, by CRISPR/Cas9 editing. Related to Fig. 1.

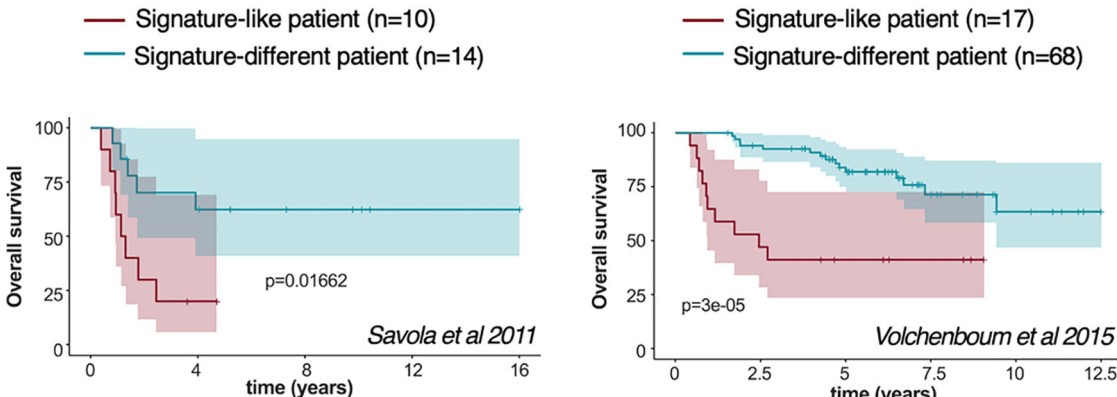

**Figure EV2. Contribution of 232 STAG2-dependent genes to survival.**

Overall survival probability (expressed as percentage) of patients from two different cohorts according to the survival signature described in main text. Only data from primary tumors were used. *P* values calculated with Cox proportional hazards regression. Related to Fig. 2.

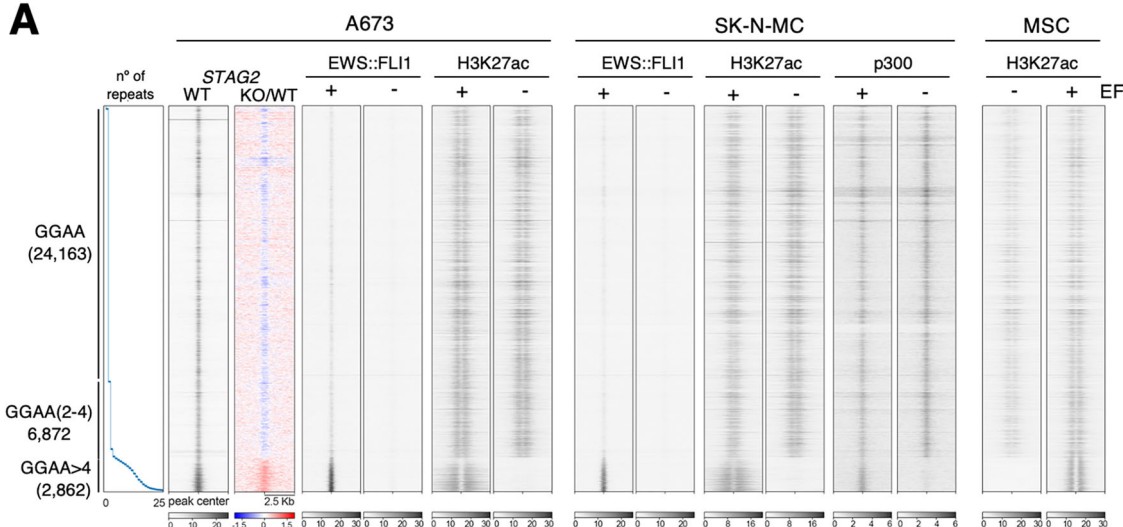

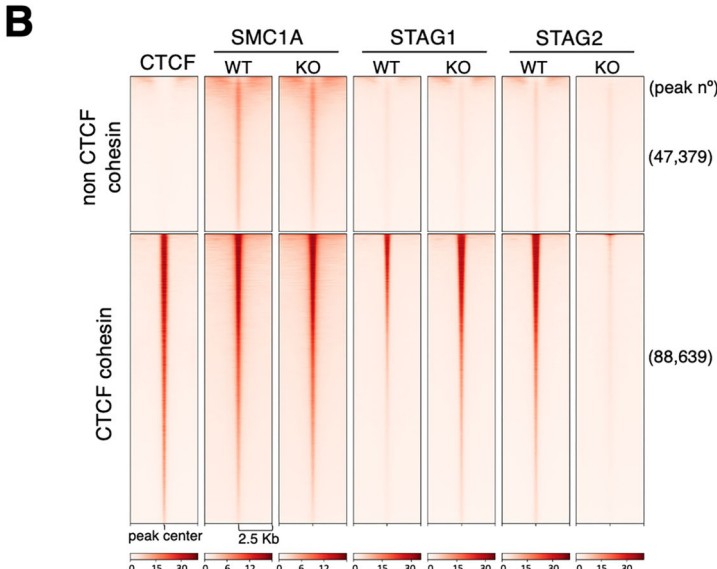

**Figure EV3. Genomic profiling in Ewing sarcoma cell lines.**

(A) Heatmaps showing EWS::FLI1 binding in A673 cells, ordered according to number of GGAA repeats. Peak calling was performed after merging data from two studies (Data ref: Surdez et al, 2021; Data ref: Adane et al, 2021). All other heatmaps showing occupancy of EWS::FLI1 (EF), H3K27ac and p300 around these peaks in A673 and SK-N-MC cells expressing shGFP (+EF) or shEF (−EF) as well as in mesenchymal stem cells (MSC) transfected with empty vector (−EF) or the oncogene (+EF), use data from (Data ref: Riggi et al, 2014). (B) Heatmaps showing ChIP-seq read distribution of cohesin in STAG2 WT or STAG2 KO A673 cells within a 5-kb window. Sites are clustered based on the presence of CTCF. Cohesin and CTCF data from (Data ref: Surdez et al, 2021) and (Data ref: Adane et al, 2021), respectively. Related to Fig. 3.

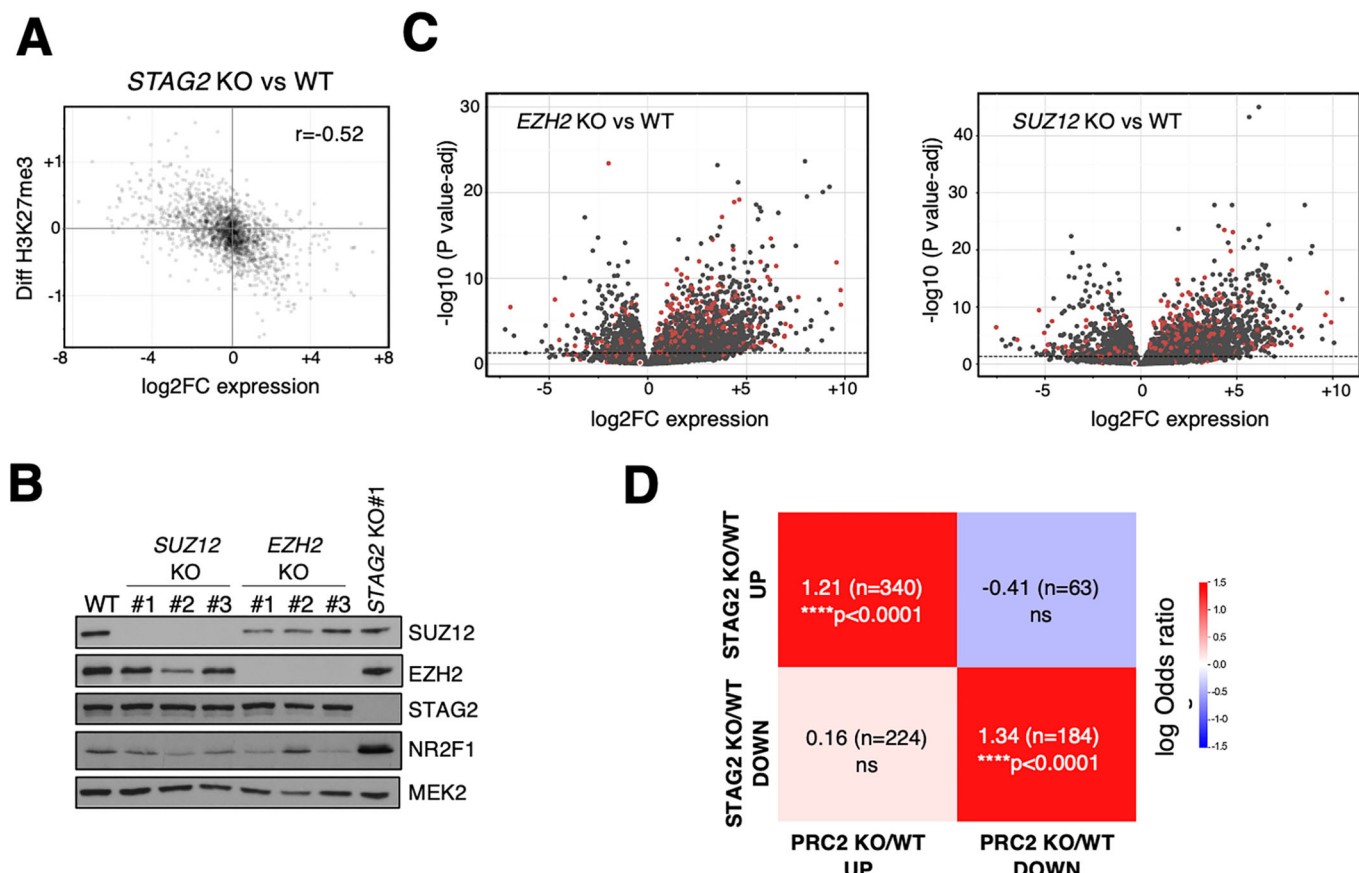

**Figure EV4. Transcriptome similarities in STAG2 KO and PRC2 KO A673 cells.**

(A) Scatterplot depicting changes in H3K27me3 at the promoter and gene expression in *STAG2* KO cells compared to STAG2 proficient cells. (B) Immunoblot analysis of whole-cell extracts of A673 clones KO for PRC2 components EZH2 and SUZ12. MEK2 serves as loading control. (C) Scatterplots showing differentially expressed genes in A673 clones KO for PRC2 components EZH2 and SUZ12. Genes deregulated also in *STAG2* KO cells (although not necessarily in the same direction) are colored in red. *P* values (*P* value-adj) were obtained using the DEseq2 package. (D) Comparison of genes significantly deregulated in PRC2 KO and *STAG2* KO A673 cells in the same direction. Chi-square test was applied. Related to Fig. 5.

## A

| Cohesin and regulators | log2FC (STAG2 KO/WT) | FDR |
|---|---|---|
| *NIPBL* | 0.176 | 0.145 |
| *MAU2* | -0.032 | 0.869 |
| *WAPL* | 0.194 | 0.12 |
| *PDS5A* | 0.133 | 0.302 |
| *PDS5B* | 0.662 | 5.44E-13 |
| *ESCO1* | 0.283 | 0.319 |
| *ESCO2* | 0.445 | 0.332 |
| *HDAC8* | 0.229 | 0.046 |
| *CTCF* | 0.147 | 0.185 |
| ***STAG1*** | 0.412 | 0.036 |
| ***STAG2*** | -1.946 | 2.84E-09 |
| ***SMC1A*** | 0.127 | 0.561 |
| ***SMC3*** | 0.092 | 0.492 |
| ***RAD21*** | -0.085 | 0.595 |

## B

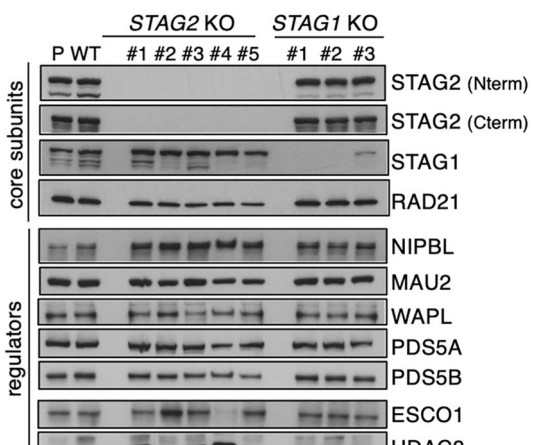

**Figure EV5. Changes in gene expression and protein abundance of cohesin subunits and regulators in A673 cells with and without STAG2.**

(A) Changes in gene expression levels of cohesin subunits and regulators assessed by RNA-seq of *STAG2* WT and KO A673 cells. Data taken from Dataset EV1A. Red and blue values correspond to significant up- and down-regulation, respectively (FDR < 0.05). (B) Immunoblot analysis of whole-cell extracts of parental (P) A673 cells and indicated clones prepared in RIPA buffer. Four gels were loaded to allow for immunoblotting with antibodies for cohesin subunits and regulators. The upper part of the figure is the same as in Fig. EV1B. Related to Fig. 7.

