## [Peer Review File · EMBO Reports]

STAG2 loss in Ewing sarcoma alters enhancer-promoter contacts dependent and independent of EWS::FLI1

Ana Losada, Daniel Giménez-Llorente, Ana Cuadrado, Maria José Andreu, Inmaculada Sanclemente-Alamán, Maria Solé-Ferran, and Miriam Rodríguez-Corsino

Corresponding author(s): Ana Losada (alosada@cniio.es) , Ana Cuadrado (acuadrado@cniio.es)

Review Timeline:

Submission Date:	17th Jul 24
Editorial Decision:	23rd Jul 24
Revision Received:	31st Jul 24
Editorial Decision:	16th Sep 24
Revision Received:	20th Sep 24
Accepted:	10th Oct 24

Editor: Deniz Senyilmaz Tiebe

Transaction Report: This manuscript was transferred to EMBO reports following peer review at The EMBO Journal.

Referee #1:

This manuscript describes studies designed to determine the mechanism through which inactivating mutations of the STAG2 tumor suppressor gene contributes to the pathogenesis of Ewing sarcoma (ES), a pediatric bone and soft tissue tumor. This is an important question since STAG2 mutations are common in ES, and an understanding of the mechanism might point in the direction of new therapies for this tumor. Unfortunately, the manuscript is entirely descriptive and fails to distinguish between direct and indirect effects of STAG2 inactivation. That said, the authors perform modern bioinformatics analysis of the effect of STAG2 inactivation in Ewing sarcoma and integrate it with other types of publicly available data to generate hypotheses about the mechanistic role of STAG2 loss in ES. This is interesting and publishable in a more specialized journal.

1) The major problem with this paper is that it is basically 100% descriptive - the claims made lack experimental evidence of causation. The authors, like others before them, use CRISPR to KO STAG2 in human ES cell lines and then perform gene expression and chromatin genomics experiments to evaluate the effects of STAG2 inactivation. They also use integrative bioinformatics to integrate other publicly available data into their analysis. While these kinds of analysis are certainly informative, there are no experiments performed to determine which (if any) of the gene expression and/or chromatin changes are actually important in STAG2 tumor suppression in Ewing sarcoma. This is particularly problematic because of #2 below.

2) The authors appear to assume that the chromatin and gene expression changes they uncover are direct effects of STAG2 inactivation, while in the vast majority of cases this is probably untrue. Since the experimental system used by the authors is stable, long term inactivation of STAG2 via CRISPR, it is basically a certainty that the vast majority of the gene expression and chromatin loop changes they uncover are indirect, downstream effects of STAG2 KO (since there has been weeks/months of culture for lots of indirect effects to accumulate).

3) One of many examples of #1 above is exemplified by text chosen for the title. The title is "Cohesin/NIPBL imbalance rewires E-P contacts..." In the manuscript the authors do show that STAG2 KO does affect the amount and types of cohesin subunits on chromatin AND they also show effects on E-P contacts. However, they never show that these two things are mechanistically related. Despite this, the title states that one causes the other. Similarly, there is substantial problematic text in the abstract stating causation without experimental evidence of it. "...Changes in CTCF-dependent chromatin contacts, unrelated to EWS::FLI1 binding, also CONTRIBUTE to the aggressive phenotype. STAG1 is unable to compensate for STAG2 loss and chromatin-bound cohesin is severely decreased, while levels of the processivity factor NIPBL remain unchanged, RESULTING IN altered DNA looping dynamics." In both cases denoted by the words in all caps, the authors provide no experimental evidence of causation yet claim causation in the abstract.

4) In an additional effort to imply causation without experimental evidence, the authors use the term "rewires" throughout the manuscript, even though the term has no scientific meaning other than "changes." However, it implies (without evidence) that these chromatin loop changes are

mechanistically meaningful in a way that the word "changes" does not.

5) The paper starts off with work to define an expression signature for STAG2 mutation in ES. This feels unrelated to the rest of the manuscript. Also, while interesting, it is not entirely clear what kind of scientific or clinical insight it provides. Furthermore, while the authors attempt to define a subset of the signature that denotes good and poor prognosis, these types of efforts require both a training set (to define a potentially predictive gene set) and a validation set (to test it) since retrospectively choosing a predictive gene set is an inherently biased activity by its nature. Unfortunately there is no validation set.

Referee #2:

The STAG2 protein is frequently absent in Ewing sarcoma samples due to mutations in the gene but also in the absence of detectible genetic alterations in STAG2. Ewing sarcoma is most commonly linked to a translocation that leads to the EWS:FLI1 oncogene, a transcription factor that can form de-novo enhancer at GGA repeat elements.

The link between STAG2-loss and changes of CTCF-dependent loops has been established already by Surdez et al. 2021, as well as an attenuating effect of STAG2 depletion on EWS:FLI1 activity under STAG2 depletion in Ewing Sarcoma cells.

The authors of the current study wanted to further understand the role of STAG2 deficiency for Ewing sarcoma and

1. Analyzed available gene expression data from an Ewing sarcoma cohort plus cell lines to identify a gene signature associated with STAG2-loss and a negative prognosis.
2. They wanted to understand which mechanism drives the misexpression of those genes.

First, the authors analyzed gene expression data from a cohort of Ewing sarcoma patients with and without mutation in STAG2 present to identify genes that depend on STAG2 and are EWS:FLI1 targets. These genes appear to be predictive for an unfortunate patient outcome and are addressed as "survival signature".

Second, the authors focus on GGAA repeats and analyze in data from the Ewing sarcoma cell line (A673) the binding of cohesin subunits and long-range interaction changes in different chromatin interaction maps (HiC, SMC1 Hi-ChIP, H3K27ac Hi-ChIP). They observe that some genes with long GGAA repeats as enhancers have lost the promoter enhancer contacts.

Subsequently, the correlation between polycomb targets and STAG2-dependent genes is investigated. Some upregulated genes after STAG2 depletions appear to be independent from polycomb. One gene, NRF2, is investigated in detail and the authors observe by promoter-capture Hi-C de-novo contacts of the promoter with a GGAA-repeat and conclude that the longer loops generated in the absence of STAG2 lead to novel EWS:FLI1 targets.

Finally, the levels of chromatin-bound proteins in the absence of STAG2 is analyzed, revealing a reduction of chromatin-bound cohesin while NIPBL levels and STAG1 levels are increased, in total extracts and SMC1 IP's.

Overall, the authors conclude that the loss of STAG2 affects the EWS:FLI1-dependent gene expression pattern by altering promoter-enhancer communication. This is an interesting new mechanistic insight that might have relevance for other cancer types since STAG2-loss is quite abundant in cancers. However, some more data would be needed to support this.

The authors demonstrated this for one example, but is this a general principle that applies to more STAG2-dependent genes, in particular in the "survival signature"?

Also, it would be important to demonstrate that the GGAA repeats that gain contacts are indeed efficient enhancers.

The authors aim to show that a EWS:FLI1-bound GGAA repeat becomes a neo enhancer for NR2F1 in the absence of STAG2 (Figure 5). To support this they perform kd of EWS:FLI1 in STAG2 ko cells. To demonstrate the dependence on STAG2 the experiment should include cells with normal STAG2 expression, e.g. the parental cells of the ko cells.

Fig.7D misses a legend

Referee #3:

Giménez-Llorente et al. investigate the consequences of STAG2 loss in Ewing sarcoma, focusing on how cohesin subunit imbalances rewire enhancer-promoter contacts and affect gene regulation. The authors demonstrate that STAG2 mutations, prevalent in this malignancy, lead to altered chromatin architecture by shifting the dynamic balance from STAG2-containing cohesin complexes to those containing STAG1, which cannot compensate functionally and disrupts normal chromatin looping. Although this isn't entirely novel as the authors' have published on this point very recently (Alonso-Gil et al., 2023), they now add that this shift impacts the regulation of genes involved in tumor progression and metastasis. Despite these significant findings, the direct functional impacts of these chromatin structural changes on cellular phenotypes associated with tumor aggressiveness is not worked out in this report. Overall, this work represents an important contribution to the field of cancer biology, particularly in the context of chromatin dynamics and its role in oncogenic transcriptional regulation. However, my evaluation is tempered by several missing statistical tests and some arguments that could be clearer. Specific comments and questions to enhance the manuscript's conclusions and readability are detailed below.

1. The argument that SA2 levels predict cancer survival is contradicted by the data presented in Figure 1A-C. WT* patients exhibit lower survival despite having SA2 levels comparable to half of the WT* cohort and generally higher than those in the mutant group. This discrepancy suggests that SA2 levels may not be directly causal to survival outcomes as proposed. Additionally, a t-test may not be suitable here if the WT* population distribution is non-normal, possibly binary. A different statistical approach or a clearer justification for the chosen method would strengthen this section.

2. In figure 2B, the authors claim that 232 "survival genes" are predictive of survival and show

survival probabilities between 109 new signature-like and non-signature-like patients. I could not find the methods for how this modeling was conducted and how this graph was generated.

3. The criteria and analytical methods used to select the 65 predictor genes from the initial set are not specified. Providing a detailed explanation of this selection process would clarify how these genes were determined to be significant.

4. I found the data and conclusions from figure 3 difficult to follow and apologize for any confusion. However, I was surprised that, as shown in 3A, SA2-KO leads to a gain of EWS:FLI1 binding at the long GGAA repeats. Specifically, SA2 KO shows a loss of SMC1 Hi-ChIP contacts at long GGAA repeats and increased EWS:FLI1 binding. But, in 3D, EWS:FLI1 KD leads to lost H3K27ac Hi-ChIP contacts at these long GGAA. How can loss of SA2 lead to gained EWS:FLI1 at these sites and lost contacts BUT EWS:FLI1 KD leads to the same loss of contact? Please clarify what EWS:FLI1 binding has to do with the contacts emerging from these repeats. Are there any non- EWS:FLI1 bound GGAA repeats to center the data on to test if SA2 could be working independently of EWS:FLI1 binding?

5. There is a major lack of statistics in all the violin plots in Figure 4 despite comparisons written between groups.

6. In Figure 4A, there is a shift in the distribution of loop sizes (lost long range & gained short range loops), which is contradictory with later figures 6-7 that discuss how SA2 loss leads to gains in long loops and lost short loops. Please clarify.

7. In Figure 4A-B, the number of contacts that are shared/gained/lost in the WT and SA2 conditions do not add up to the number of contacts shown for each condition.

8. The conclusion that these EWS:FLI1 -bound GGAA repeats are acting as enhancers is not supported by any evidence in this manuscript. This should be reworded to tone down the speculation.

9. The significance of Polycomb in the context of the gene signature associated with SA2 loss is ambiguous, especially given the ratio of upregulated to downregulated genes. Expanding on the role and implications of Polycomb in this setting, or providing additional data to support its relevance, would clarify its place in the study's broader conclusions.

10. In figure 6, the stats are again missing from A-C

Referee #4:

Giminez-Llorente present a multi-omic analysis of critical regulators in Ewing Sarcoma (EWS), i.e. EWS:FLI1 and STAG2 (cohesin). By re-analysing existing data they find that some STAG2 WT EWS samples phenocopy STAG2 mutant EWS samples indicating loss at the transcriptional or (post-

)translational level. Furthermore, they construct a signature that is associated with improved survival, which is related to the STAG2 status. Next they use EWS cell lines that establish the role of STAG2 in EWS:FLI1 regulation. They find that sites with > 4 GGAA sequences (the binding motif of EWS::FLI1) and that are bound by cohesin show strong a strong stripe behavior (indicative of loop extrusion). Knock-down of EWS::FLI1 diminishes these stripes (however, this is expected since this is shown by doing a H3K27ac Hi-ChIP and Figure S3A shows a near complete loss of H3K27ac at these sites).

The author describe three examples of genes that are affected by STAG2 loss (ADRA1D, NR2F1 and RNF141) using Promoter Capture Hi-C. Finally the authors show how loss of STAG2 affects the association of cohesin subunits and regulators to chromatin in EWS cells.

The paper presents interesting data and the results seem solid. The data support the conclusions. The paper is trying to perform a functional analysis of genes that are misregulated by EWS::FLI1 and STAG2 in EWS and to provide mechanistic insight into how these factors achieve this. This makes the manuscript difficult to follow at times. The analysis in Figure 7 is interesting, because it provides an explanation how with less cohesin on chromatin STAG2 KO cells do manage to have longer loops (i.e. the NIPBL/PDS5 ratio and less chance of cohesin complexes bumping into each other during extrusion).

The only major comment I have is that the authors should show the actual data for the Promoter Capture Hi-C data. This way the reader can assess the impact on the contact frequency in general, rather than the highly abstract arcs, which are always a bit difficult to interpret. This pertains to Figure 4G, 5A and 6D.

Other comments:

Figure 2D: is the enrichment of EWS::FLI1 targets significant? Without knowing the background frequency this is hard to estimate. Please perform a hypergeometric test to determine this.

Figure 3C: can these results also be shown with APAs?

Related: please explain in more detail how the analyses in Figure 3B-D are performed and what they are showing. Not all readers will be familiar with a stripe analysis.

Figure 3D: "consistent with the observed increase in H3K27ac-mediated interactions (Fig. 3D)" I do not see the interaction increase in this figure.

Figure S5B: please present scatterplots of the RNAseq data of the EZH2 and SUZ12 KO cells (of the log2FC data). The text indicates that transcriptome data is shown in this figure but only Western blots are shown.

Figure 7A: "A less noticeable decrease in chromatin was observed for WAPL and PDS5A" I see a very clear decrease in WAPL levels. The WAPL levels seems as much decreased as SMC1A.

Dear Ana,

Thank you for transferring your manuscript to EMBO Reports, which was previously reviewed at another venue.

Having read the manuscript and the referee reports, I would like to invite you to submit a revised manuscript to EMBO Reports as previously communicated, where the text indicating causality of the findings is toned down (as per referee #1, point 3). Moreover, all other concerns/recommendations of referees #2, #3 and #4 need to be addressed.

Please address all referee concerns in a complete point-by-point response. Acceptance of the manuscript will depend on a positive outcome of a second round of review. It is EMBO reports policy to allow a single round of major experimental revision only and acceptance or rejection of the manuscript will therefore depend on the completeness of your responses included in the next, final version of the manuscript.

We realize that it is difficult to revise to a specific deadline. In the interest of protecting the conceptual advance provided by the work, we recommend a revision within 3 months. Please discuss the revision progress ahead of this time with me if you require more time to complete the revisions, or if you have questions or comments regarding the revision (also by video chat).

1. A data availability section providing access to data deposited in public databases is missing (where applicable).
2. Your manuscript contains statistics and error bars based on $n=2$. Please use scatter plots in these cases.

You can submit the revision either as a Scientific Report or as a Research Article. For Scientific Reports, the revised manuscript can contain up to 5 main figures and 5 Expanded View figures, and it should not exceed 27000 characters. If the revision leads to a manuscript with more than 5 main figures it will be published as a Research Article. In this case the Results and Discussion section should be separate. If a Scientific Report is submitted, these sections have to be combined. This will help to shorten the manuscript text by eliminating some redundancy that is inevitable when discussing the same experiments twice. In either case, all materials and methods should be included in the main manuscript file.

4) a .docx formatted letter INCLUDING the reviewers' reports and your detailed point-by-point responses to their comments. As part of the EMBO publication's Transparent Editorial Process, EMBO reports publishes online a Review Process File (RPF) to accompany accepted manuscripts. This File will be published in conjunction with your paper and will include the referee reports, your point-by-point response and all pertinent correspondence relating to the manuscript.

<https://www.embopress.org/page/journal/14693178/authorguide#transparentprocess>

5) a complete author checklist, which you can download from our author guidelines <https://www.embopress.org/page/journal/14693178/authorguide>. Please insert information in the checklist that is also reflected in the manuscript. The completed author checklist will also be part of the RPF.

6) Please note that all corresponding authors are required to supply an ORCID ID for their name upon submission of a revised manuscript (<<https://orcid.org/>>). Please find instructions on how to link your ORCID ID to your account in our manuscript tracking system in our Author guidelines <<https://www.embopress.org/page/journal/14693178/authorguide#authorshipguidelines>>

Additional information on source data and instruction on how to label the files are available: <https://www.embopress.org/page/journal/14693178/authorguide#sourcedata>

9) Our journal encourages inclusion of *data citations in the reference list* to directly cite datasets that were re-used and obtained from public databases. Data citations in the article text are distinct from normal bibliographical citations and should directly link to the database records from which the data can be accessed. In the main text, data citations are formatted as follows: "Data ref: Smith et al, 2001" or "Data ref: NCBI Sequence Read Archive PRJNA342805, 2017". In the Reference list, data citations must be labeled with "[DATASET]". A data reference must provide the database name, accession number/identifiers and a resolvable link to the landing page from which the data can be accessed at the end of the reference. Further instructions are available at <http://www.embopress.org/page/journal/14693178/authorguide#referencesformat>

12) Please also note our reference format: <http://www.embopress.org/page/journal/14693178/authorguide#referencesformat>

13) All Materials and Methods need to be described in the main text using our 'Structured Methods' format, which is required for

all research articles. According to this format, the Methods section includes a Reagents and Tools Table (listing key reagents, experimental models, software and relevant equipment and including their sources and relevant identifiers) followed by a Methods and Protocols section describing the methods using a step-by-step protocol format. The aim is to facilitate adoption of the methodologies across labs. More information on how to adhere to this format as well as a downloadable template (.docx) for the Reagents and Tools Table can be found in our author guidelines:
<https://www.embopress.org/page/journal/14693178/authorguide#structuredmethods>.

An example of a Method paper with Structured Methods can be found here:
<https://www.embopress.org/doi/10.15252/msb.20178071>.

I look forward to seeing a revised version of your manuscript when it is ready. Please let me know if you have questions or comments regarding the revision.

Kind regards,

Deniz

Deniz Senyilmaz Tiebe, PhD
Scientific Editor
EMBO Reports

Please find below our **point-by point response** to referees. We thank them for their comments, which have been addressed in the revised version of the manuscript.

Reviewer #1:

This manuscript describes studies designed to determine the mechanism through which inactivating mutations of the STAG2 tumor suppressor gene contributes to the pathogenesis of Ewing sarcoma (ES), a pediatric bone and soft tissue tumor. This is an important question since STAG2 mutations are common in ES, and an understanding of the mechanism might point in the direction of new therapies for this tumor. Unfortunately, the manuscript is entirely descriptive and fails to distinguish between direct and indirect effects of STAG2 inactivation. That said, the authors perform modern bioinformatics analysis of the effect of STAG2 inactivation in Ewing sarcoma and integrate it with other types of publicly available data to generate hypotheses about the mechanistic role of STAG2 loss in ES. This is interesting and publishable in a more specialized journal.

1) The major problem with this paper is that it is basically 100% descriptive - the claims made lack experimental evidence of causation. The authors, like others before them, use CRISPR to KO STAG2 in human ES cell lines and then perform gene expression and chromatin genomics experiments to evaluate the effects of STAG2 inactivation. They also use integrative bioinformatics to integrate other publicly available data into their analysis. While these kinds of analysis are certainly informative, there are no experiments performed to determine which (if any) of the gene expression and/or chromatin changes are actually important in STAG2 tumor suppression in Ewing sarcoma. This is particularly problematic because of #2 below.

We respectfully disagree with this criticism. We obtained a STAG2-dependent gene signature in Ewing sarcoma using data from cell lines and patients with and without STAG2. With data from a different cohort, we then showed that this gene signature predicts prognosis. Next, we performed experiments in Ewing sarcoma cells with and without STAG2 that identify changes in the chromatin interactome that can explain, at least in part, the transcriptomic changes. Furthermore, we provide an explanation for the altered contacts based on the analyses of relative amounts of cohesin and its regulators. All these data are functionally relevant and unveil the consequences of STAG2 depletion in the transcriptome of Ewing sarcoma cells. The reviewer is right, however, when he/she says that we do not address the "cellular" consequences of these gene expression changes and their contribution to tumorigenesis.

2) The authors appear to assume that the chromatin and gene expression changes they uncover are direct effects of STAG2 inactivation, while in the vast majority of cases this is probably untrue. Since the experimental system used by the authors is stable, long term inactivation of STAG2 via CRISPR, it is basically a certainty that the vast majority of the gene expression and chromatin loop changes they uncover are indirect, downstream effects of STAG2 KO (since there has been weeks/months of culture for lots of indirect effects to accumulate).

We again disagree with the reviewer. For all experiments, cells have been in culture for a few passages (after generation of the KO clones). We have used five different STAG2 KO clones to identify common changes. Moreover, please notice that the STAG2-dependent 232 genes also become deregulated in cells knocked down for STAG2 by siRNA treatment for 72h (see Figure 2A, Fig. S2A in previous version). While we agree with the reviewer that some of these changes may be indirect, we also identify changes in chromatin contacts upon KO of STAG2 that are likely to drive (or contribute to) gene expression changes.

3) One of many examples of #1 above is exemplified by text chosen for the title. The title is "Cohesin/NIPBL imbalance rewires E-P contacts..." In the manuscript the authors do show that STAG2 KO does affect the amount and types of cohesin subunits on chromatin AND they also show effects on E-P contacts. However, they never show that these two things are

mechanistically related. Despite this, the title states that one causes the other. Similarly, there is substantial problematic text in the abstract stating causation without experimental evidence of it. "...Changes in CTCF-dependent chromatin contacts, unrelated to EWS::FLI1 binding, also CONTRIBUTE to the aggressive phenotype. STAG1 is unable to compensate for STAG2 loss and chromatin-bound cohesin is severely decreased, while levels of the processivity factor NIPBL remain unchanged, RESULTING IN altered DNA looping dynamics." In both cases denoted by the words in all caps, the authors provide no experimental evidence of causation yet claim causation in the abstract.

We have toned down the statements related to causality in title, abstract and throughout main text.

New title: STAG2 loss in Ewing sarcoma changes E-P contacts dependent and independent of EWS::FLI1

In Abstract: "Changes in CTCF-dependent chromatin contacts involving signature genes, unrelated to EWS::FLI1 binding, ~~also contribute to the aggressive phenotype~~ were also identified. STAG1 is unable to compensate for STAG2 loss and chromatin-bound cohesin is severely decreased, while levels of the processivity factor NIPBL remain unchanged, ~~resulting in altered~~ likely affecting DNA looping dynamics. These results illuminate how STAG2 loss ~~rewires~~ modifies the chromatin interactome of Ewing sarcoma cells ~~to promote metastasis~~ and provide a list of potential biomarkers and therapeutic targets."

Also, in Introduction: "Thus, complete loss of STAG2 may alter the relative amount of cohesin and its regulators and impact loop extrusion dynamics, ~~thereby changing the transcription of genes that are relevant for disease progression and patient survival~~ with consequences for the transcriptome."

4) In an additional effort to imply causation without experimental evidence, the authors use the term "rewires" throughout the manuscript, even though the term has no scientific meaning other than "changes." However, it implies (without evidence) that these chromatin loop changes are mechanistically meaningful in a way that the word "changes" does not.

We have replaced or eliminated the term from title and abstract.

5) The paper starts off with work to define an expression signature for STAG2 mutation in ES. This feels unrelated to the rest of the manuscript. Also, while interesting, it is not entirely clear what kind of scientific or clinical insight it provides. Furthermore, while the authors attempt to define a subset of the signature that denotes good and poor prognosis, these types of efforts require both a training set (to define a potentially predictive gene set) and a validation set (to test it) since retrospectively choosing a predictive gene set is an inherently biased activity by its nature. Unfortunately there is no validation set.

We apologize for the lack of clarity. We selected 232 STAG2-dependent genes based on transcriptomic data from a first patient cohort (reported in Tirode 2014). We then found that STAG2 WT* patients and STAG2 MUT patients have both bad prognosis compared to STAG2 WT patients and similar expression for these genes, suggesting a correlation with prognosis. To test this possibility, we then used data from the primary tumors of two additional, independent cohorts (Savola 2011; Volchenboum 2015) either merged or separated (Figure 2B and S2B, respectively, in original manuscript), to demonstrate that the 232-gene signature has prognostic value. The use of this 232-gene signature may identify cases with poor prognosis in the absence of mutations in STAG2.

The selection of a subset of genes (65) within these 232 as those that are more important for survival is based on the fact that each gene, individually, separates patients with good or bad prognosis with a p value <0.05 (Table S1C in original manuscript). We agree with the reviewer

that the prognostic value of the 65-gene “reduced survival signature” would require additional validation. Since we do not have at the moment these additional data, we omit the 65-gene subset in the revised manuscript, and refer only to the 232 gene set as “survival signature”. We have removed Figure 2D and instead transfer Figure S2A of the original manuscript to main Figure 2A to show how the 232 genes are expressed in the patients of the first cohort (and how a subset of patients without STAG2 mutations (STAG2 WT*) cluster with the STAG2 MUT patients).

Reviewer #2:

The STAG2 protein is frequently absent in Ewing sarcoma samples due to mutations in the gene but also in the absence of detectible genetic alterations in STAG2. Ewing sarcoma is most commonly linked to a translocation the lead to the EWS:FLI1 oncogene, a transcription factor that can form de-novo enhancer at GGA repeat elements.

The link between STAG2-loss and changes of CTCF-dependent loops has been established already by Surdez et al. 2021, as well as a attenuating effect of STAG2 depletion on EWS:FLI1 activity under STAG2 depletion in Ewing Sarcoma cells.

The authors of the current study wanted to further understand the role of STAG2 deficiency for Ewing sarcoma and

- 1. Analyzed available gene expression data from an Ewing sarcoma cohort plus cell lines to identify a gene signature associated with STAG2-loss and a negative prognosis.*
- 2. They wanted to understand which mechanism drives the misexpression of those genes.*

First, the authors analyzed gene expression data from a cohort of Ewing sarcoma patients with and without mutation in STAG2 present to identify genes that depend on STAG2 and are EWS:FLI1 targets. These genes appear to be predictive for an unfortunate patient outcome and are addressed as "survival signature".

Second, the authors focus on GGAA repeats and analyze in data from the Ewing sarcoma cell line (A673) the binding of cohesin subunits and long-range interaction changes in different chromatin interaction maps (HiC, SMC1 Hi-ChIP, H3K27ac Hi-ChIP). They observe that some genes with long GGAA repeats as enhancers have lost the promoter enhancer contacts.

Subsequently, the correlation between polycomb targets and STAG2-dependent genes is investigated. Some upregulated genes after STAG2 depletions appear to be independent from polycomb. One gene, NRF2, is investigated in detail and the authors observe by promoter-capture Hi-C de-novo contacts of the promoter with a GGAA-repeat and conclude that the longer loops generated in the absence of STAG2 lead to novel EWS:FLI1 targets.

Finally, the levels of chromatin-bound proteins in the absence of STAG2 is analyzed, revealing a reduction of chromatin-bound cohesin while NIPBL levels and STAG1 levels are increased, in total extracts and SMC1 IP's.

Overall, the authors conclude that the loss of STAG2 affects the EWS:FLI1-dependent gene expression pattern by altering promoter-enhancer communication. This is an interesting new mechanistic insight that might have relevance for other cancer types since STAG2-loss is quite abundant in cancers. However, some more data would be needed to support this.

The authors demonstrated this for one example, but is this a general principle that applies to more STAG2-dependent genes, in particular in the "survival signature"?

Figures 4H, 6E and Appendix Fig. S1 (previously Fig. S4) present Venn diagrams highlighting which genes of the survival signature can be potentially explained by changes in E-P contacts detected by PC-HiC in STAG2 KO cells. In order to address a comment from reviewer#1, we now omit the 65-gene reduced survival signature and refer only to the 232-gene signature. Venn diagrams have been modified accordingly.

Also, it would be important to demonstrate that the GGAA repeats that gain contacts are indeed efficient enhancers.

GGAA>4 motifs have been proposed to behave as neoenhancers, as shown in the data from Riggi et al (2014) replotted in Fig. EV3A (Fig. S3A in previous version). These sites have H3K27acetylation and P300 in Ewing sarcoma cells but both are lost after knock down of the oncogene. Moreover, these sites acquire H3K27 acetylation when mesenchymal stem cells are transfected with the oncogene. The GGAA>4 repeat that establishes a new contact with the *NR2F1* promoter in *STAG2* KO cells does show the active H3K27acetylation mark already in *STAG2* WT cells, and this mark further increases in the *STAG2* KO, supporting its role as an enhancer (this can be seen in the ChIP-seq tracks in Figure 5A). If we plot H3K27acetylation mark in the fraction of EWS::FLI1-bound GGAA>4 repeats that gain contacts in *STAG2* KO cells (81 sites), we observe a similar behavior. This does not demonstrate, but suggests, that they are enhancers.

The authors aim to show that a EWS:FLI1-bound GGAA repeat becomes a neo enhancer for *NR2F1* in the absence of *STAG2* (Figure 5). To support this they perform kd of EWS:FLI1 in *STAG2* ko cells. To demonstrate the dependence on *STAG2* the experiment should include cells with normal *STAG2* expression, e.g. the parental cells of the ko cells.

In *STAG2* proficient cells, there is little expression of *NR2F1* (see the RNA-seq expression data in the lower part of Fig. 5A and the low expression and protein accumulation in Parental (P), *STAG2* proficient cells in Fig. 5B). Depletion of the oncogene would not show much of an effect. In contrast, we show that *NR2F1* becomes expressed in these cells (both Parental and *STAG2* WT cells) after downregulation of CTCF (Fig. 5C), consistent with our hypothesis that a CTCF site blocks the contact between *NR2F1* promoter and the upstream long GGAAA repeat.

Fig.7D misses a legend

The panels were wrongly labeled. We have corrected this mistake.

Reviewer #3:

Giménez-Llorente et al. investigate the consequences of *STAG2* loss in Ewing sarcoma, focusing on how cohesin subunit imbalances rewire enhancer-promoter contacts and affect gene regulation. The authors demonstrate that *STAG2* mutations, prevalent in this malignancy, lead to altered chromatin architecture by shifting the dynamic balance from *STAG2*-containing cohesin complexes to those containing *STAG1*, which cannot compensate functionally and disrupts normal chromatin looping. Although this isn't entirely novel as the authors' have published on this point very recently (Alonso-Gil et al., 2023), they now add that this shift impacts the regulation of genes involved in tumor progression and metastasis. Despite these significant findings, the direct functional impacts of these chromatin structural changes on cellular phenotypes associated with

tumor aggressiveness is not worked out in this report. Overall, this work represents an important contribution to the field of cancer biology, particularly in the context of chromatin dynamics and its role in oncogenic transcriptional regulation. However, my evaluation is tempered by several missing statistical tests and some arguments that could be clearer. Specific comments and questions to enhance the manuscript's conclusions and readability are detailed below.

1. The argument that SA2 levels predict cancer survival is contradicted by the data presented in Figure 1A-C. WT patients exhibit lower survival despite having SA2 levels comparable to half of the WT* cohort and generally higher than those in the mutant group. This discrepancy suggests that SA2 levels may not be directly causal to survival outcomes as proposed. Additionally, a t-test may not be suitable here if the WT* population distribution is non-normal, possibly binary. A different statistical approach or a clearer justification for the chosen method would strengthen this section.*

We did not mean to imply that STAG2 expression levels predict survival. In fact, in Discussion we write that "... loss of STAG2 staining in patient biopsies of primary tumors is a better predictor of outcome than finding mutations or reduced STAG2 expression".

However, we acknowledge that the sentence that we wrote in the original manuscript is misleading: We have therefore changed the sentence and the order of panels B and C in Fig. 1.

Before revision: STAG2 WT* tumors tended to have lower STAG2 mRNA levels than the rest of non-mutant or wild-type (WT) STAG2 cases (Fig. 1B) and a worse prognosis (Fig. 1C).

Revised: STAG2 WT* tumors had worse prognosis than non-mutant or wild-type (WT) STAG2 cases (Fig. 1B) while their STAG2 mRNA levels were not necessarily lower (Fig. 1C).

Also, following the reviewer's comment, we have applied a Wilcoxon test to the data in (now) Figure 1C.

2. In figure 2B, the authors claim that 232 "survival genes" are predictive of survival and show survival probabilities between 109 new signature-like and non-signature-like patients. I could not find the methods for how this modeling was conducted and how this graph was generated.

This analysis is described in the last paragraph of the Methods section ("Analysis of patient cohorts"). We have added one sentence at the end (underlined) that we feel was missing and changed the mention to Figure panels according to the revisions made:

"To classify patients in "signature-like" and signature-different" expression, we used the 232 genes that were differentially expressed in patients and in A673 clones. Initially, a "STAG2 KO model" was constructed, wherein each upregulated gene had the maximum value in the dataset for that gene, while each downregulated gene had the minimum value. This model represents an extreme phenotype associated with STAG2 loss. Conversely, a "STAG2 WT model" was established using the opposite rationale: for each upregulated gene, the model had the minimum value, and *vice versa* for downregulated genes. Subsequently, unsupervised clustering (k-means) with two groups and centroids initialized to STAG2 WT and STAG2 KO model values was employed to categorize patients of two patient cohorts (Savola et al., 2011; Volchenboum et al., 2015) as either "signature-like" or "signature-different". Finally, PCA segregated patients according to the previously determined clusters (Fig. 2B). Overall survival probabilities were calculated for patients in each group. This unbiased analysis was performed independently in the two cohorts and after merging the two datasets using the 'sva' R package to mitigate biases stemming from different datasets (Appendix Fig. S1 and Fig. 2C, respectively).

Also, in the PCA in current Fig. 2B, we referred to the models as "standard" (denoted by asterisks). We now use "model", to follow the same nomenclature used in Methods.

3. The criteria and analytical methods used to select the 65 predictor genes from the initial set are not specified. Providing a detailed explanation of this selection process would clarify how

these genes were determined to be significant.

The selection of a subset of 65 genes was based on the fact that each gene, individually, separates patients with good or bad prognosis with a p value <0.05 (indicated in the last row of previous Table S1C). However, to address a criticism from reviewer#1 we have decided to eliminate the mention to this subset and instead refer to the 232 STAG2-dependent genes as the survival signature.

4. I found the data and conclusions from figure 3 difficult to follow and apologize for any confusion. However, I was surprised that, as shown in 3A, SA2-KO leads to a gain of EWS:FLI1 binding at the long GGAA repeats. Specifically, SA2 KO shows a loss of SMC1 Hi-ChIP contacts at long GGAA repeats and increased EWS:FLI1 binding. But, in 3D, EWS:FLI1 KD leads to lost H3K27ac Hi-ChIP contacts at these long GGAA. How can loss of SA2 lead to gained EWS:FLI1 at these sites and lost contacts BUT EWS:FLI1 KD leads to the same loss of contact? Please clarify what EWS:FLI1 binding has to do with the contacts emerging from these repeats. Are there any non- EWS:FLI1 bound GGAA repeats to center the data on to test if SA2 could be working independently of EWS:FLI1 binding?

We apologize for the lack of clarity. Increased binding of the oncoprotein to GGAA long repeats does not ensure increased looping if cohesin STAG2 is not present. In other words, both EWS:FLI1 binding and cohesin-STAG2 are required for looping. We have clarified this in the text (underlined):

“SMC1 Hi-ChIP data revealed a clear decrease in these interactions in STAG2 KO cells, suggesting that cohesin-STAG1 cannot compensate for the loss of STAG2 and proper contacts with distant regions cannot be properly established despite increased binding of the oncoprotein to long GGAAA repeats (Fig. 3C).

5. There is a major lack of statistics in all the violin plots in Figure 4 despite comparisons written between groups.

The reviewer is right. In the box plots we did not perform any test to compare the different “groups”. We have now corrected this and used Wilcoxon test in Figures 4B and 4C and Kruskal-Wallis test for the data in 4D and 4E.

6. In Figure 4A, there is a shift in the distribution of loop sizes (lost long range & gained short range loops), which is contradictory with later figures 6-7 that discuss how SA2 loss leads to gains in long loops and lost short loops. Please clarify.

Our interpretation of the graph in Figure 4A is that there is a decrease in loops of 80-800 kb in size, the usual size for cohesin-mediated loops. The number of loops longer than that is small (128 for loops > 1 Mb in Figure 6A) and therefore difficult to appreciate in the graph of Figure 4A. As for the increase in smaller loops (below 100 kb), they are most likely not mediated by cohesin and probably not included in Figure 6A. We have added a sentence in the description of this Figure in main text (see next comment). We have also changed the graph to improve visualization of overlapped curves.

7. In Figure 4A-B, the number of contacts that are shared/gained/lost in the WT and SA2 conditions do not add up to the number of contacts shown for each condition.

The reviewer is right. We called loops individually for the four cell lines/clones (Parental, WT, STAG2 KO#1, STAG2 KO#2), with the requirement of at least five reads in the interaction and a score >3, resulting in more than 200,000 loops called in each cell line/clone (Figure 4A). Then, we compared the loops called in the two cell lines/clones of each condition: 54,226 loops were detected in both the Parental and WT cells, whereas 37,142 were detected in the two STAG2 null clones. To classify the loops in common, gained or lost in Figure 4B, we established the following criteria: loops called only in the STAG2 KO condition (in both clones) were classified as

“gained”, those present only in the STAG2 WT condition (in both WT and Parental cells) were classified as “lost” while “common” loops had to be present in at least 3 cell line/clones (P, WT and one STAG2 KO clone or both STAG2 KO clones and either P or WT). Thus, if a loop is called in P, WT and KO#1 but not in KO#2, it will be considered “common”, but it is not among the 37,142 loops present in both KO. Likewise, a loop present in P or WT and in the two KO clones, is also “common” but it is not among the 54,226 loops in “WT condition”. This is why numbers do not add up.

We have clarified this in the Methods section (in which the description was, in fact, not entirely correct). We have modified what we say in main text and removed the numbers from Figure 4A (but not from main text). We have also added a sentence in the same paragraph referred to the previous point made by the reviewer:

Before revision

The loop length distribution profiles were very similar for the two biological replicates of each genetic condition, with the STAG2 deficient clones presenting a reduced number of interactions in a slightly wider loop size range (Fig. 4A). We ended up with 54,226 and 37,142 bona-fide interactions called in STAG2 WT and STAG2 KO cells, respectively. Of these, 43,339 were present in both conditions (“common”), 11,309 arose (“gained”) and 22,102 disappeared (“lost”) in STAG2 deficient cells (Fig. 4B, Table S4). Importantly, “common” interactions tended to be weaker in these cells than in STAG2 proficient cells.

Revised:

The loop length distribution profiles were very similar for the two biological replicates of each genetic condition, with the STAG2 deficient clones presenting a reduced number of interactions in the size range expected for cohesin-mediated loops (80-800 Kb; Fig. 4A). To select the most robust interactions, we restricted subsequent analyses to contacts that were present in both STAG2 proficient cell lines (Parental and WT clone, 54,226 loops) or in both STAG2 KO clones (37,142 loops). Among these contacts, we considered “common” loops those that were present in three out of four cell lines while “gained” loops were only present in the STAG2 KO clones and “lost” loops were only called in STAG2 expressing cells (Fig. 4B, Dataset EV2.)

8. The conclusion that these EWS:FLI1 -bound GGAA repeats are acting as enhancers is not supported by any evidence in this manuscript. This should be reworded to tone down the speculation.

The proposal that GGAA long repeats act as neoenhancers was formulated by Riggi et al., 2014 and is commonly found in the literature thereafter (e.g. Tomazou et al., 2015; Sheffeld et al., 2017). Data from the first study are used in Fig. EV3A (previously Fig. S3A)and show that the oncoprotein binds more strongly to long ($n>4$) GGAA repeats, and this binding is essential for their activity as enhancers, marked by the presence of H3K27ac and P300.

Nevertheless, we have addressed the reviewer request in the revised manuscript, for instance:

“A heatmap of the EWS::FLI1 peaks ordered according to the number of GGAA repeats they encompass shows that the oncoprotein binds more strongly to long ($n>4$) GGAA repeats, also considered microsatellites, and this binding is essential for their proposed activity as enhancers (Fig. EV3A)”.

9. The significance of Polycomb in the context of the gene signature associated with SA2 loss is ambiguous, especially given the ratio of upregulated to downregulated genes. Expanding on the role and implications of Polycomb in this setting, or providing additional data to support its relevance, would clarify its place in the study's broader conclusions.

We agree with the reviewer. Unfortunately, we have been unable to find a robust link between STAG2 and Polycomb in the Ewing sarcoma cells. We generated Polycomb (PRC2) KO clones in A673 to produce a list of Polycomb direct targets as those that become deregulated in EZH2

KO and SUZ12 KO cells- Out of such list of genes, only a fraction respond similarly to loss of PRC2 or loss of STAG2 and we could not identify distinct features for this gene subset (e.g., changes in chromatin contacts with H3K27me3 regions, size of compartment or position within a compartment).

To address a request from reviewer 4, we have now included in current Fig. EV4 (previous Fig. S5) the scatterplots of the RNAseq data of the EZH2 KO and SUZ12 KO cells and added also information about which genes become deregulated in STAG2 KO cells (not necessarily in the same direction): Figure EV4C.

10. In figure 6, the stats are again missing from A-C

These have been added to the new version of the manuscript.

Reviewer #4:

Giminez-Llorente present a multi-omic analysis of critical regulators in Ewing Sarcoma (EWS), i.e. EWS::FLI1 and STAG2 (cohesin). By re-analysing existing data they find that some STAG2 WT EWS samples phenocopy STAG2 mutant EWS samples indicating loss at the transcriptional or (post-)translational level. Furthermore, they construct a signature that is associated with improved survival, which is related to the STAG2 status. Next they use EWS cell lines that establish the role of STAG2 in EWS::FLI1 regulation. They find that sites with > 4 GGAA sequences (the binding motif of EWS::FLI1) and that are bound by cohesin show strong a strong stripe behavior (indicative of loop extrusion). Knock-down of EWS::FLI1 diminishes these stripes (however, this is expected since this is shown by doing a H3K27ac Hi-ChIP and Figure S3A shows a near complete loss of H3K27ac at these sites).

We thank the reviewer for this comment. We have now replaced the metaplots generated with the H3K27ac Hi-ChIP data (in Figure 3D) with metaplots generated with available Hi-C data of A673 cells before and after EWS::FLI1 KD (Shownpil et al 2002, ref 7). The decrease in interactions established from GGAA long repeats is clearly seen.

The author describe three examples of genes that are affected by STAG2 loss (ADRA1D, NR2F1 and RNF141) using Promoter Capture Hi-C. Finally the authors show how loss of STAG2 affects the association of cohesin subunits and regulators to chromatin in EWS cells.

The paper presents interesting data and the results seem solid. The data support the conclusions. The paper is trying to perform a functional analysis of genes that are misregulated by EWS::FLI1 and STAG2 in EWS and to provide mechanistic insight into how these factors achieve this. This makes the manuscript difficult to follow at times. The analysis in Figure 7 is interesting, because it provides an explanation how with less cohesin on chromatin STAG2 KO cells do manage to have longer loops (i.e. the NIPBL/PDS5 ratio and less chance of cohesin complexes bumping into each other during extrusion).

The only major comment I have is that the authors should show the actual data for the Promoter Capture Hi-C data. This way the reader can assess the impact on the contact frequency in general, rather than the highly abstract arcs, which are always a bit difficult to interpret. This pertains to Figure 4G, 5A and 6D.

We do not quite understand this criticism. Arcs are the most common way to present significant chromatin interactions between distal regions obtained in capture Hi-C data. In the figures presented in main text, for clarity, we only represent contacts emanating from the genes under study but the session in the genome browser accessible to reviewers (<https://genome-euro.ucsc.edu/s/Dinamica%20cromosomica/Reviewers%20session%20EWS>) includes tracks with all significant contacts in cells with and without STAG2. The genomic coordinates of these loops are listed in Dataset EV2 (previously Table S4) together with their strength and

classification. We have now added information about the differential interactions. Raw data are uploaded in GEO.

Nevertheless, we have prepared for the reviewer a version of Figure 4G in which we have added tracks with a representation similar to the one used for 4C data, considering the promoter of the gene as “viewpoint”. We do not think that this add much information, and prefer not to include it in the manuscript.

Other comments:

Figure 2D: is the enrichment of EWS::FLI1 targets significant? Without knowing the background frequency this is hard to estimate. Please perform a hypergeometric test to determine this.

To address a criticism of reviewer#1, we have eliminated Figure 2D and now refer to the 232 STAG2-dependent genes as survival signature. Among them, there are 55 genes that become deregulated upon knock down of EWS::FLI1 (according to RNA-seq data in studies by Tomazou 2015 and Surdez 2021). Applying a hypergeometric test, p value of enrichment of EWS::FLI1 targets among these 232 STAG2 deregulated genes is 0.0003394797. However, we wish not to highlight this fraction, on the contrary, that there are many transcriptional changes independent of EWS::FLI1 related with patient prognosis.

In Dataset EV1C (previous Table S1C), we list the 232 STAG2 dependent genes and shadow the 55 genes that are also deregulated in EWS::FLI1 KD A673 cells.

We have also modified Table S5 (Dataset EV3 in current version) to include all EWS::FLI1 target genes according to those RNA-seq data and indicate the ones that we have considered “direct” targets based on ChIP-seq and PC-HiC data.

Figure 3C: can these results also be shown with APAs?

Related: please explain in more detail how the analyses in Figure 3B-D are performed and what they are showing. Not all readers will be familiar with a stripe analysis.

We have a couple of reasons for choosing stripe analysis over Aggregate Peak Analysis (APA). The main one is probably that we wish to analyze the loop extrusion process. The other is that the number of called loops to be used in APA is much lower than the number of interactions plotted in our metaplots. Just to show the reviewer, we have carried out the APA using loops called in our PC-HiC experiment and compared the signal intensity in STAG2 WT and STAG2 KO for the cohesin containing loops between promoters and genomic regions containing GGAA (1-4) or GGAA>4. Despite the small number of loops called with GGAA>4 regions (n=375), a slight decrease in intensity is observed in the absence of STAG2 specifically in this type of loops.

We now also included additional information of the metaplots in the corresponding Figure legend: “Metaplots that aggregate chromatin interactions emanating from the indicated features (GGAA repeats of different length, with or without cohesin) and extending up to 0.5 Mb away in both directions. Numbers (n, below each metaplot) in (C) and (D) are the same as in (B) and come from the analyses shown in (A). Color scales represent the ratio of Observed over Expected interactions (log2). Datasets used in Table EV1.”

Figure 3D: "consistent with the observed increase in H3K27ac-mediated interactions (Fig. 3D)" I do not see the interaction increase in this figure.

The reviewer is right, the increase in H3K27ac-mediated interactions was difficult to see in the metaplot as presented. However, after replacement of the Figure 3D, now it is more clearly seen that Hi-C detected interactions emanating from single or short repeats do increase after oncogene KD.

Figure S5B : please present scatterplots of the RNAseq data of the EZH2 and SUZ12 KO cells (of the log2FC data). The text indicates that transcriptome data is shown in this figure but only Western blots are shown.

We have now included scatterplots of the RNAseq data of the EZH2 and SUZ12 KO cells, as requested.

Figure 7A: "A less noticeable decrease in chromatin was observed for WAPL and PDS5A" I see a very clear decrease in WAPL levels. The WAPL levels seems as much decreased as SMC1A.

The reviewer is right, we have modified the text to indicate so:

“A similar decrease in chromatin was observed for WAPL while the reduction in PDS5A was less noticeable.”

Dear Ana,

Thank you for submitting your revised manuscript. It has now been seen by two of the original referees.

My apologies for the delay in getting back to you. It took longer than anticipated to receive the referee reports.

As you can see, the referee finds that the study is significantly improved during revision and recommends publication. However, I need you to address the points below before I can accept the manuscript.

- Please address the remaining concerns of the referees. I concur with referee #2 regarding the title.
- Please provide 3-5 keywords for your study. These will be visible in the html version of the paper and on PubMed and will help increase the discoverability of your work.
- Please rename the 'Conflict of interest' section as "Disclosure Statement and Competing Interests".
- We note the following name discrepancy - Daniel Giménez-Llorente in the manuscript vs. Daniel Giménez in manuscript tracking system.
- As per our format requirements, in the reference list, citations should be listed in alphabetical order and then chronologically, with the authors' surnames and initials inverted; where there are more than 10 authors on a paper, 10 will be listed, followed by 'et al.'. Please see <https://www.embopress.org/page/journal/14693178/authorguide#referencesformat>
- We note that there is one preprint in the reference list (Olmedo-Pelayo et al., 2023). Citations to manuscripts posted on recognized preprint servers can be cited the following way:

In-text citation: (preprint: NAME1 et al, YEAR)

Author NAME1, Author NAME2, (YEAR) article title. bioRxiv doi: 1234/002.djf123 [PREPRINT]

- We note the following regarding funding information: missing in the manuscript tracking system: AEI/10.13039/501100011033, Ministerio de Ciencia, Innovación y Universidades, the European Regional Development Fund (ERDF-EU), CNIO Friends
- We note that Fig. 7D is called out in the text, but panel D is not labeled in the figure.
- Please upload Reagents & Tools table separately by choosing the relevant file type.
- I believe that Table EV3 should be a part of the Reagents & Tools table.
- The word "Appendix" is missing in the figure label in the Appendix file (should be Appendix Figure S1).
- Materials and Methods section should be called Methods.
- The following paragraphs need to be removed from the ms: "Extended View Datasets" and "Appendix Supplementary Figures"
- Please make the dataset GSE267223 publicly available, remove the reviewer token from the manuscript, and add a link that directly resolves to the dataset.
- Our production/data editors have asked you to clarify several points in the figure legends:
 - o Please note that the exact p values are not provided in the legends of figures 4b-f; 5b-c; 6a-c; EV 4d.
 - o Please indicate the statistical test used for data analysis in the legends of figures 1d; 2d; 5b-c; EV 4c-d.
 - o Please note that the box plots need to be defined in terms of minima, maxima, centre, bounds of box and whiskers, and percentile in the legends of figures 2d; 4b-f; 6a-c.
 - o Please note that information related to n is missing in the legends of figures 2d; 5b-c.
 - o Although 'n' is provided, please describe the nature of entity for 'n' in the legends of figures 4b-f; 6a, c.
 - o Please note that the error bars are not defined in the legend of figure 5c.
 - o Please note that for heatmap present in figure 1d, a numbered scale bar is not provided. This needs to be rectified.
 - o Please note that the colored dots are not defined in the legend of figure 7b. This needs to be rectified.
- Papers published in EMBO Reports include a 'synopsis' and 'bullet points' to further enhance discoverability. Both are displayed on the html version of the paper and are freely accessible to all readers. The synopsis includes a short standfirst summarizing the study in 1 or 2 sentences (max 35 words) that summarize the paper and are provided by the authors and streamlined by the handling editor. I would therefore ask you to include your synopsis blurb and 3-5 bullet points listing the key experimental findings.
- In addition, please provide an image for the synopsis. This image should provide a rapid overview of the question addressed in the study but still needs to be kept fairly modest since the image size cannot exceed 550 (width) x 300-600 (height) pixels.

Thank you again for giving us to consider your manuscript for EMBO Reports, I look forward to your minor revision.

Kind regards,

Deniz

--

Deniz Senyilmaz Tiebe, PhD
Senior Scientific Editor
EMBO Reports

Referee #1:

The authors have done a great job of answering my questions.

However, they do not agree with my main point of critique, which is that it is important to show the quantitative information for the Capture-C data. When I look at the figure they provide in the rebuttal I believe it is important information. The authors state that arcs are the most common way of representing 3D genome interactions. That may be so, but that does not make them useful visualizations of the data. The plot they present shows five interactions. Two of those seem meaningful based on the quantitative data. Three are more difficult to interpret. It is important for the reader to know this. Therefore I would strongly urge the authors to include the quantitative data as Supp. Figures for all the regions that they show in the paper.

Another point that would need clarification is the Hi-C data they added in Figure 3B-D. An important result with respect to the main message of the paper is in Figure 3B & D. According to the legend these are the same data. I do not understand why there is such a big difference between for instance the GGAA > 4 control + cohesin in B and D if this is based on the same data. Please explain where this difference comes from and make it clear what the difference is in the legend and text.

Referee #2:

The authors have significantly revised the manuscript. Overall, the quality of the manuscript has increased since claims by the authors match now better with the actual data.

I find this manuscript acceptable for publication in EMBO reports.

I apologize for taking so long. The omission of figure numbers on the figures made cross-referencing between texts and figures a challenge.

Points to revise:

I would strongly suggest to the authors to come up with a better title. The abbreviation E-P is not suitable for a title.

The current Figure 2C would actually indicate a better overall survival of "signature-like" patients with survival bit lower than 75% over 16 years. The authors should revise this figure carefully.

On page 7: "Among the 232 STAG2-dependent genes only 55 are targets of EWS::FLI1, that is, they become deregulated after oncogene KD (shadowed in Dataset EV1C). We conclude that STAG2 loss in Ewing sarcoma results in altered transcription beyond EWS::FLI1 target genes that most likely contribute to adverse prognosis."

These sentences should be revised to be more to the point.

Figure 7 misses a labeling of the y-axis.

Response to referees

Referee #1:

The authors have done a great job of answering my questions.

However, they do not agree with my main point of critique, which is that it is important to show the quantitative information for the Capture-C data. When I look at the figure they provide in the rebuttal I believe it is important information. The authors state that arcs are the most common way of representing 3D genome interactions. That may be so, but that does not make them useful visualizations of the data. The plot they present shows five interactions. Two of those seem meaningful based on the quantitative data. Three are more difficult to interpret. It is important for the reader to know this. Therefore I would strongly urge the authors to include the quantitative data as Supp. Figures for all the regions that they show in the paper.

We have now added Appendix Figures S1, S3 and S4 to provide the quantitative information regarding contacts from the gene promoter corresponding to the regions shown in Figures 4G, 5A and 6D, respectively, as requested by the reviewer

Another point that would need clarification is the Hi-C data they added in Figure 3B-D. An important result with respect to the main message of the paper is in Figure 3B & D. According to the legend these are the same data. I do not understand why there is such a big difference between for instance the GGAA > 4 control + cohesin in B and D if this is based on the same data. Please explain where this difference comes from and make it clear what the difference is in the legend and text.

The reviewer is right. Data used for metaplots came from two different studies, with different sequencing depth of the Hi-C experiments (for those in Figure 3B we used data from Sanalkumar et al., 2023 (GSE193824) while for the metaplots in 3D, we used Showpnil et al. 2022 (GSE185125) which has less valid reads. This was a mistake, and we have now used the data from Sanalkumar et al., 2023 (GSE193824) both in 3B and 3D metaplots. In the final version of Figure 3D we only show the metaplots for the EF KD condition, as those in the control condition are already shown in 3B.

Referee #2:

The authors have significantly revised the manuscript. Overall, the quality of the manuscript has increased since claims by the authors match now better with the actual data.

I find this manuscript acceptable for publication in EMBO reports.

I apologize for taking so long. The omission of figure numbers on the figures made cross-referencing between texts and figures a challenge.

Points to revise:

I would strongly suggest to the authors to come up with a better title. The abbreviation E-P is not suitable for a title.

We have changed the title to "STAG2 loss in Ewing sarcoma alters enhancer-promoter contacts dependent and independent of EWS::FLI1"

The current Figure 2C would actually indicate a better overall survival of "signature-like" patients with survival bit lower than 75% over 16 years. The authors should revise this figure carefully.

Thank you for pointing out this mistake: the graph legends were swapped! This has been corrected in Figure 2C and Figure EV2.

On page 7: "Among the 232 STAG2-dependent genes only 55 are targets of EWS::FLI1, that is,

they become deregulated after oncogene KD (shadowed in Dataset EV1C). We conclude that STAG2 loss in Ewing sarcoma results in altered transcription beyond EWS::FLI1 target genes that most likely contribute to adverse prognosis."
These sentences should be revised to be more to the point.

We have rewritten the sentence: "Out of the 232 STAG2-dependent genes, only 55 are also targets of EWS::FLI1, that is, they become deregulated after oncogene KD (highlighted in Dataset EV1C). We conclude that STAG2 loss in Ewing sarcoma leads to transcriptional changes that extend beyond EWS::FLI1 target genes and that most likely contribute to adverse prognosis."

Figure 7 misses a labeling of the y-axis.
The label has been added.

Additional changes made to the manuscript

- The 'Conflict of interest' section has been renamed "Disclosure Statement and Competing Interests".
- Citation format has been updated. Citations for custom-made antibodies (now listed in Reagents & tools table) have been added to the Reference list.
- Information previously included in Table EV3 (Oligonucleotides and Antibodies) is now part of the Reagents & Tools table. A short section "Antibodies" has been included in Methods, as description of a new custom made antibody was missing in the original draft.
- The following paragraphs have been removed from: "Extended View Datasets" and "Appendix Supplementary Figures"
- Exact p values have been included either in the legend or in the graphs when $p > 0.0001$
- Figure legends have been revised to ensure that statistical tests are indicated, to define boxplots, to add information about "n" and to define error bars.
- A link to the GEO database is provided.

Dr. Ana Losada
Spanish National Cancer Research Centre (CNIO)
Molecular Oncology Programme
Melchor Fernandez Almagro 3
Madrid E-28029
Spain

Dear Ana,

Thank you for submitting your revised manuscript. I have now looked at everything and all is fine. Therefore, I am very pleased to accept your manuscript for publication in EMBO Reports.

Congratulations on a nice work!

There is one minor point to address before we can export your manuscript to our publishers. I note that the manuscript contains data citations (i.e. GSE132966 - Surdez et al., 2021, Tomazou et al., 2015, ICGC BOCA-FR - Tirode et al., 2014, GSE17618 - Savola et al., 2011, GSE63157 - Volchenboum et al., 2015, GSE116495 - Adane et al., 2021, GSE133154 - Surdez et al., 2021, GSE61953 - Riggi et al., 2014, GSE193824 - Sanalkumar et al., 2023, GSE165977 - Adane et al., 2021), whose in-text and reference list citation formats need to be updated similar to the below example.

Hörnberg E, Ylitalo EB, Crnalic S, Antti H, Stattin P, Widmark A, Bergh A, Wikström P (2011) Gene Expression Omnibus GSE29650 (<https://www.ncbi.nlm.nih.gov/geo/query/acc.cgi?acc=GSE29650>). [DATASET]

Hörnberg E, Ylitalo EB, Crnalic S, Antti H, Stattin P, Widmark A, Bergh A, Wikström P (2011) Expression of androgen receptor splice variants in prostate cancer bone metastases is associated with castration-resistance and short survival. PLoS One 6: e19059

In the main text, these datasets should be cited with the prefix "Data ref:" to distinguish them from the reference to the original article that reported the dataset. Example:

"...were grouped based on the relative levels of AR-Vs expressed, mainly AR-V7 (Hörnberg et al, 2011; Data ref: Hörnberg et al, 2011)."

Please see <https://www.embopress.org/page/journal/14693178/authorguide#referencesformat> for further information.

You can send the updated manuscript text file per email. Thank you.

Kind regards,

Deniz

--

Deniz Senyilmaz Tiebe, PhD
Senior Scientific Editor
EMBO Reports
